cognition/psychology

exploration and exploitation, decision making, problem solving, heuristic

**Author for correspondence:**
Kyanoush Seyed Yahosseini
e-mail: yahosseini@mpib-berlin.mpg.de

# Search as a simple take-the-best heuristic

## Kyanoush Seyed Yahosseini and Mehdi Moussaïd

Center for Adaptive Rationality, Max Planck Institute for Human Development, Berlin, Germany

KSY, 0000-0003-0094-5971; MM, 0000-0002-1268-6332

Humans commonly engage in a variety of search behaviours, for example when looking for an object, a partner, information or a solution to a complex problem. The success or failure of a search strategy crucially depends on the structure of the environment and the constraints it imposes on the individuals. Here, we focus on environments in which individuals have to explore the solution space gradually and where their reward is determined by one unique solution they choose to exploit. This type of environment has been relatively overlooked in the past despite being relevant to numerous real-life situations, such as spatial search and various problem-solving tasks. By means of a dedicated experimental design, we show that the search behaviour of experimental participants can be well described by a simple heuristic model. Both in rich and poor solution spaces, a take-the-best procedure that ignores all but one cue at a time is capable of reproducing a diversity of observed behavioural patterns. Our approach, therefore, sheds lights on the possible cognitive mechanisms involved in human search.

## 1. Introduction

Many aspects of life can be seen as search for rewarding outcomes [1,2]. Animals forage for food or mates [3,4], people search for information in the Internet or for a smart move during a chess game [5] and organizations search for new market opportunities [6]. Search involves at least two main components: (i) sampling solutions by *exploring* the environment, and (ii) collecting rewards by *exploiting* the discovered solutions [1,2,6]. For example, a hungry tourist searching for a place to eat in a foreign city will first examine the surrounding streets and evaluate the quality of several restaurants (exploration), before eventually deciding which one to have dinner at (exploitation).

Exploration and exploitation are not always mutually exclusive [2,7]. In fact, numerous search problems exhibit a structure where the pay-off of every explored solution is immediately earned and accumulates over time. This situation typically induces a dilemma between exploring new solutions and exploiting known ones [8]. For example, in multi-armed bandit problems (MABs) a gambler

needs to decide which of the many slot machines to play next, while simultaneously collecting the reward of each decision [8,9]. Similarly, foraging hummingbirds combine feeding on nectar with the pursuit of finding new flowers [2]. In other types of search problems, however, exploration and exploitation are *temporally separated*. In these cases, as in the hungry tourist example, the reward is based on one single solution that the individual chooses by stopping the exploration process and transitioning from exploration to exploitation. This type of search problems has been studied for example in the sampling paradigm [10] or in the secretary problem [11].

Another dimension along which search problems differ is how the environment can be explored [12]. In some situations, individuals can directly move between two distant solutions, irrespective of their location. That is, they can *jump* between remote regions of the solution space, without the necessity to move through all the intermediate solutions. Examples include search for information on the Internet, where individuals are free to switch from one website to any other, or Mason and Watts' network experiment where participants harvest resources in a virtual landscape [13]. In other search problems, however, jumps between distant solutions are not possible. In such problems, the exploration is *gradual*, that is, it is constrained to the neighbourhood of the current solution. Our hungry tourist, for example, can only evaluate the quality of the adjacent restaurants. Likewise, people searching for a good solution to the travelling salesman problem in the experiment performed by Dry *et al.* [14] can only add or remove one connection at a time, exploring the solution space gradually.

These two features of the search problem (i.e. the separation of exploration and exploitation and the gradual exploration) can have a strong impact on the way individuals deal with search problems. Yet, research has mostly focused on search problems where exploration and exploitation happen simultaneously (e.g. MAB [15]; abstract search [16]; Lévy processes [17]; comparison of different paradigms [18]) and/or where jumps in the solution space are allowed (e.g. correlated MAB [19]; sampling paradigm [10]; secretary problem [20]; random sampling [21]). Nevertheless, many search problems are characterized by separated exploration and exploitation phases and gradual exploration; examples include animals deciding where to hunt prey [22,23], algorithms maximizing their reward in a reinforcement learning settings [24] and humans visually searching for a lost item [25] or solving a complex problem [5,14]. How do people solve such search tasks?

In the present work, we focus on such type of problems by addressing two research questions:

(1) How do people explore their environment when only gradual movements are possible (i.e. jumps are not feasible)?
(2) When do people decide to terminate the exploration and start exploiting one solution when these two phases are temporally separated?

We address these two questions separately by means of a dedicated experimental design. In a first experimental phase, we study specifically how people explore a two-dimensional solution space when only gradual movements are allowed. In a second experimental phase, we study the decision to exploit a solution, when exploration and exploitation are separated. We show that these two behavioural components are well described by a simple model based on the take-the-best heuristic (TTB) [26,27]. A third experimental phase, combining exploration and exploitation, confirms that the full model has captured the participants' search strategies, both in rich and poor solution spaces. We conclude by comparing the model to alternative approaches proposed in the literature.

# 2. Material and methods

We conducted an experiment in which participants were instructed to search for the best possible solution in a *landscape*—a conceptual Euclidean representation of a solution space [13,28,29]. In a landscape, each field represented one solution and was associated with one fixed pay-off. Participants were given 30 rounds of search. In each round, they could move to one of the neighbouring solutions or stay at their current one. Participants indicated their decision by clicking on the solution they wanted to move to (see electronic supplementary material, figure A1 for the experimental interface). Movements to distant solutions that were not adjacent to the currently occupied one were not allowed (gradual exploration). Participants saw the pay-offs associated with their current solution and to the eight adjacent ones. We call *trajectory* the sequence of solutions a participant moved through during the 30 rounds of the experiment. The experiment was divided into three phases (see also table 1):

(i) *The exploration phase.* In the first phase, participants were positioned in the centre of a squared landscape containing $63 \times 63$ fields. The landscape was large enough to ensure that participants could not reach its border within the allocated time. Participants were instructed to search for

**Table 1.** Description of the three experimental phases. Every phase consisted of 20 landscapes (divided into 10 rich and 10 poor landscapes) played for 30 rounds each.

| phase | landscape | size | start | reward |
| --- | --- | --- | --- | --- |
| exploration | two-dimensional | $63 \times 63$ | (32, 32) | highest pay-off |
| exploitation | uni-dimensional | $1 \times 63$ | (1) | pay-off in last round |
| combined | two-dimensional | $63 \times 63$ | (32, 32) | pay-off in last round |

the best possible solution and were rewarded based on the highest pay-off they found after 30 rounds of exploration. In such a way, we could focus solely on the exploration pattern, leaving out the exploitation decision.

(ii) *The exploitation phase.* In the second stage, participants were placed at one end of a uni-dimensional landscape, that is, a vector containing 63 solutions. They were rewarded according to the pay-off of the field they occupied in the last round. With this design, we focus on how and when the participants decide to stop exploring the solution space and start exploiting one solution.

(iii) *The combined phase.* In the third phase, participants were positioned in the middle of a quadratic landscape containing $63 \times 63$ (as in the exploration phase) and rewarded according to the pay-off of the position they occupied in the last round (as in the exploitation phase).

In each phase, participants played a total of 10 rich landscapes containing a high number of peaks, and 10 poor landscapes containing a low number of peaks. Rich and poor landscapes were presented to the participants in a random order.

## 2.1. Landscapes

The two-dimensional landscapes used in the exploration and combined phase were produced according to the following procedure:

(i) We first generated $n$ sub-landscapes. Each sub-landscape consisted of a $63 \times 63$ matrix filled with zeroes, except for one randomly selected field that contained a random value drawn from a normal distribution with mean zero and standard deviation one. We call this non-zero field a peak. We then squared the peak value to avoid negative pay-offs.

(ii) We then applied a Gaussian filter with a standard deviation of one on each sub-landscape to create a local gradient around the peak.

(iii) We finally merged the $n$ sub-landscapes into a single one by selecting the highest pay-off across all sub-landscapes at each coordinate.

We used $n = 32$ and $n = 512$ to create poor and rich landscapes, respectively. The uni-dimensional landscapes used for the exploitation phase were generated by randomly selecting one horizontal line from a two-dimensional landscape of the same type.

Finally, the pay-offs were rounded to the closest integer and linearly scaled between zero and a random value between 30 and 80. This procedure generates landscapes similar to those shown in figure 1. We call the normalized pay-off of a solution its actual pay-off divided by the highest pay-off of the landscape.

## 2.2. Procedure and participants

Participants were recruited from the Max Planck Institute for Human Development's participant pool and gave informed consent to the experiment. The experimental procedure was approved by the Ethics Committee of the Max Planck Institute for Human Development.

At the beginning of each phase, participants received information about the goal of the search, the size of the landscape, the moving rules and completed a practice search. We recruited 50 participants (28 female, mean age = 26.1, s.d. = 4.4). They received a flat fee of €12 plus a monetary bonus based on their aggregated performance (€0.15 per 100 points, mean bonus = €2.46, s.d. = €0.48). The average completion time was 35.31 min (s.d. = 10.67 min).

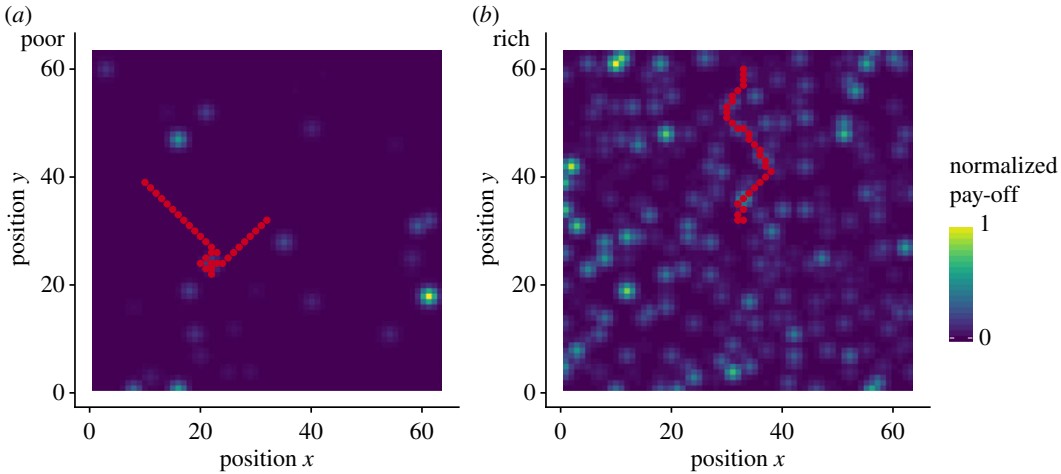

**Figure 1.** Two examples of exploration trajectories (in red) in a poor (*a*) and a rich (*b*) landscape. The red dots indicate the participants' trajectory, i.e. the spatial position at each round. The colour coding of the landscape indicates the normalized pay-off associated with each position (*x*, *y*). Participants always started in the middle of the landscape, at position (32, 32).

## 3. Results

### 3.1. Exploration phase

In the first phase of the experiment, we focused on how people explore their environment when jumps are not permitted. For this, participants were instructed to search in a two-dimensional landscape of size $63 \times 63$ and rewarded based on the highest pay-off they found during 30 rounds of search. Figure 1 shows two illustrative trajectories of participants in a poor and a rich landscape.

Overall, participants performed better in rich than in poor landscapes (the average normalized pay-off was 0.33 with s.d. = 0.21 in rich landscapes and 0.17 with s.d. = 0.26 in poor landscapes), but explored a similar number of unique solutions (on average 26.1 and 27.2 fields visited in rich and poor landscapes, respectively, with an s.d. of 6.64 and 5.39, respectively). In both environments, participants avoided revisiting previously visited solutions. On average, the fraction of solutions visited more than once was 0.14 (s.d. = 0.20). As a comparison, this fraction goes up to 0.57 (s.d. = 0.10) for a completely random exploration process, suggesting that participants are actively avoiding previously visited solutions.

To better understand the exploration mechanisms at play, we aggregated all trajectories (in poor and rich landscapes separately) to generate the corresponding density maps. These maps indicate how often each field has been visited relative to the others, irrespective of the peaks positions.

As shown in figure 2, the density map for the poor landscapes reveals a surprising X-shaped exploration pattern, suggesting that participants have a preference for diagonal movements. This pattern, however, is not visible in the rich landscapes.

Why do participants tend to explore along diagonal lines when peaks are rare but not when they are abundant? Diagonal movements uncover up to five new solutions because it enables participants to move simultaneously along the *x*- and *y*-dimensions. By contrast, vertical or horizontal moves can only reveal up to three new solutions. Therefore, moving diagonally constitutes an efficient strategy to extend one's exploration range. The absence of the X-shaped exploration pattern in rich landscapes suggests that participants might rely on another cue when peaks are frequent: the surrounding pay-off values. In this case, participants are most likely to use a hill-climbing process consisting in moving to the most-rewarding adjacent solution [30]. Overall, our data suggest that three rules are guiding the exploration: (i) not returning to previously visited solutions (using the *non-visited cue*), (ii) maximizing the immediate pay-off by moving to the most-rewarding neighbouring solution (using the *pay-off cue*), and (iii) maximizing the number of new solutions revealed (using the *novelty cue*). Figure 3 confirms the important role played by these three components. We formalize these three rules in a simple lexicographic model based on the TTB [26,27]. TTB assumes that decisions between multiple options are made by ranking cues and then looking at only one cue at a time. If a cue discriminates, a decision is made for the best option, and only otherwise the next cue is evaluated.

In our experiment, participants need to decide between nine different options in every round (that is, moving to one of the eight neighbouring solutions or staying at the current one). To make that decision,

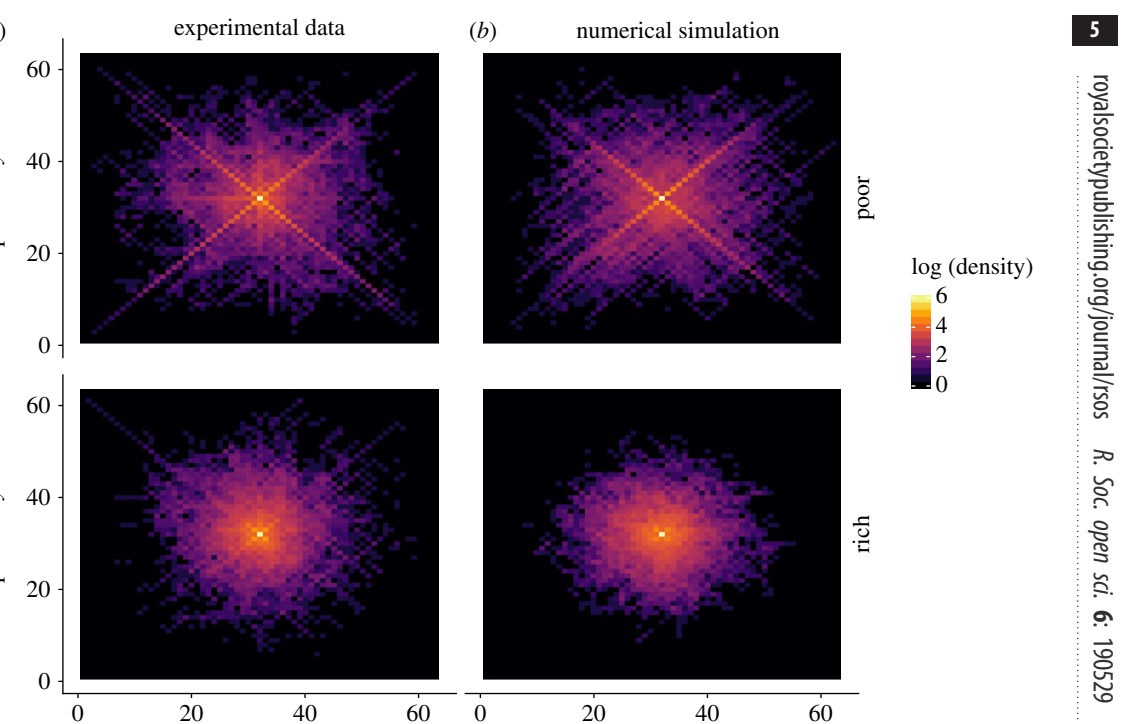

**Figure 2.** Density maps for poor (upper line) and rich (lower line) landscapes in the exploration phase, as observed in the experimental data (*a*) and obtained from numerical simulations (*b*). The colour coding indicates how often a given position (*x*, *y*) has been visited at the aggregate level, represented in the logarithmic scale. The starting point of the search is located in the middle of the map, at coordinates (32, 32). For the simulations, we randomly selected the same number of trajectories as in the behavioural data to ensure comparable density scales.

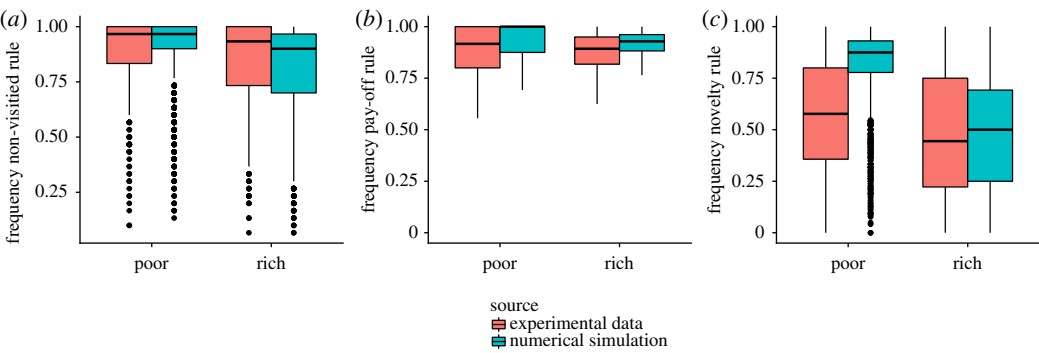

**Figure 3.** Frequency of decisions based on the three rules as observed in the experimental data and in numerical simulations (for poor and rich landscapes). Box plots indicating the interquartile range (box), the median (horizontal line) and 1.5-times interquartile range (whiskers). Outliers are shown as a single dot. (*a*) Non-visited cue. Proportion of movements towards a non-visited solution. (*b*) Pay-off cue. Proportion of movements towards a non-visited solution with the highest pay-off. (*c*) Novelty cue. Proportions of movements maximizing the number of new solutions revealed, when the non-visited cue and the pay-off cue do not discriminate. In (*c*), the decisions based on the novelty cue in poor landscapes are less frequent than predicted by the model, but remain nevertheless higher than what a random search model predicts (40%). Overall, the three cues presented in (*a*–*c*) account for 68.8% of all the decisions made by the participants.

we assume that participants first look at the not-visited cue and stop if the cue discriminates (i.e. if only one option has not been visited, that particular option is chosen and a decision is made). If the *not-visited cue* does not discriminate, they consider the *pay-off cue* of the remaining options (if exactly one option has a higher pay-off than all other options, that option is chosen and a decision is made). If the pay-off cue still does not discriminate, they finally consider the *novelty cue* (that is the option that reveals the largest number of new solutions is chosen). If two or more equivalent options remain at the end of the process, a

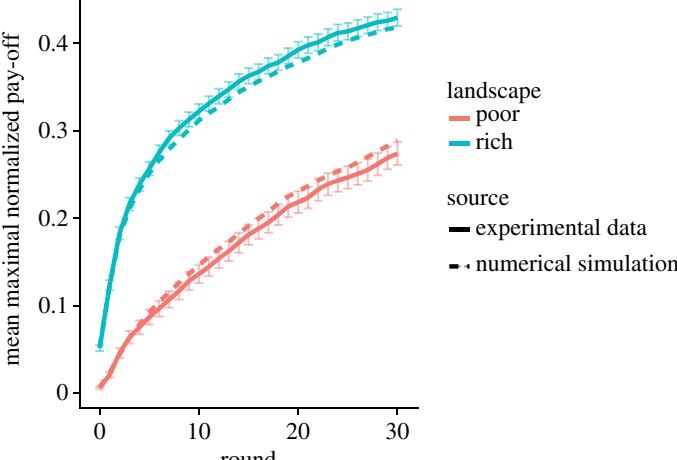

**Figure 4.** Performance in the exploration phase. The highest normalized pay-off found as a function of time averaged over all participants (solid line) or produced in simulations (dashed line). Error bars for the experimental data indicate the standard error of the mean.

**Table 2.** Example of the TTB. The decision-maker can choose between nine options (as columns: the eight cardinal directions and staying (∅)) knowing the three cue values for each option (as rows: non-visited, pay-off and novelty). In this example, the decision-maker would first look at the non-visited cue for all options. This stage leaves seven possible non-visited options that are then compared based on the pay-off cue. At this stage, one option scores better than the others (the 'East' option marked in bold letters) and a decision is made in favour of it. The novelty cue is not examined. Green cells (resp. red cells) indicate cue values that are considered (resp. ignored) during the decision.

| options<br>cues | ∅ | N | NE | **E** | SE | S | SW | W | NW |
|---|---|---|---|---|---|---|---|---|---|
| non-visited | 0 | 0 | 1 | 1 | 1 | 1 | 1 | 1 | 1 |
| pay-off | 0 | 45 | 5 | **25** | 10 | 5 | 0 | 0 | 0 |
| novelty | 0 | 0 | 2 | 3 | 5 | 3 | 5 | 3 | 2 |

random decision is made between them. Finally, we add a uniform noise parameter $\epsilon$ to the model, defined as a low probability to make a random choice between all available options. Table 2 shows an example of how decisions are made according to our model. The model has one unique parameter, namely, the noise level $\epsilon$. We fitted $\epsilon$ by systematically varying it between zero and one and comparing the highest pay-off per round to the experimental data (averaged over 8000 simulated trajectories). The value $\epsilon = 0.17$ minimizes the squared difference to the experimental observations (figure 4). We keep $\epsilon$ constant in the remainder of the study.

Once fitted to the pay-off curves, the resulting model also produces consistent patterns on other aspects of the search behaviour, such as the X-shaped exploration pattern (figure 2) and the influence of the three cues (figure 3). In the next section, we will extend this model to include the decision to exploit.

## 3.2. Exploitation phase

In the second phase of the experiment, we examine how people decide to stop their search and exploit a previously discovered solution. For this, participants are positioned at one end of a uni-dimensional landscape, and are rewarded based on the pay-off of the field they occupy at the last round. Unlike the exploration phase, participants can only navigate along a line (i.e. by moving left or right). Furthermore, they face a trade-off between the benefits of exploring as far as possible and the need to be positioned at a sufficiently good solution after 30 rounds. Figure 5 illustrates this experiment.

The type of landscape directly impacted the participants' performances (average final pay-off 0.60 in rich landscapes and 0.47 in poor landscapes, with an s.d. of 0.36 and 0.46, respectively). On average

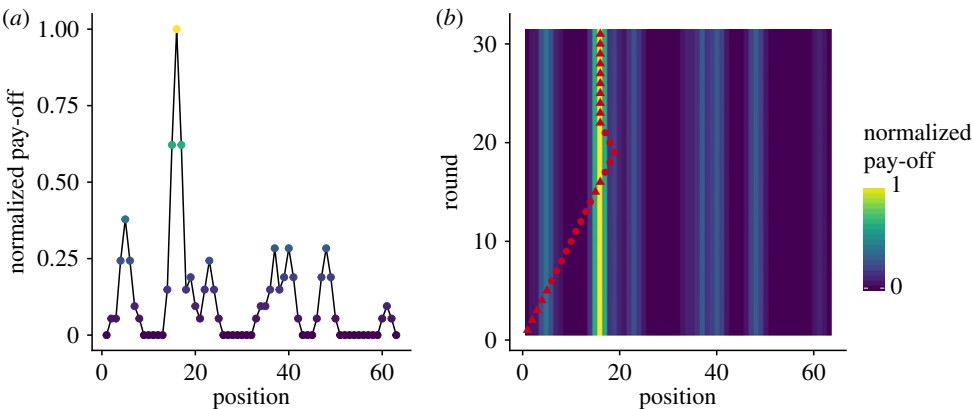

**Figure 5.** Exploitation phase. (a) Example of a rich landscape used in the experiment. Participants always started at position $x = 1$ and could see the pay-offs associated with their current position and to the two neighbouring ones. In the first round, they could only move to the right. Participants were rewarded based on the pay-off of their final position. (b) Trajectory of a participant navigating in the landscape shown in (a). The red markers indicate the position in each round and the colour coding shows the normalized pay-off of each position. In this example, the participant discovered a first peak at round 5 (at position $x = 5$) but continued her exploration. At round 15, a better solution is found (at position $x = 15$). The participant continued her exploration for 3 rounds before returning to her best solution and settling there until the end of the 30 rounds. Triangles indicate that the current solution is the best that has been discovered so far (named $X_{\text{best}}$ in the main text).

participants moved 20.6 fields (s.d. = 9.93) and 13.2 fields (s.d. = 8.71) away from their starting point in poor and rich landscapes, respectively.

The trajectory shown in figure 5b illustrates that, after discovering a new best solution, participants tend to continue searching for a more advantageous one before eventually returning back to it if no better solution is found.

At any moment of time, we call $X_{\text{best}}$ the best pay-off that the participant has discovered so far and $P_{\text{best}}$ the position of that best pay-off. We first ask the following question: How far do participants continue their exploration after discovering $P_{\text{best}}$, before coming back to it if no better one is found? From a normative point of view, participants should start returning to $P_{\text{best}}$ as late as possible, such that their chance to discover a better solution is maximized. This optimal exploration range simply equals to $D_{\text{optimal}} = \lfloor T_{\text{left}}/2 \rfloor$, where $T_{\text{left}}$ is the remaining number of rounds. In such a way, the participant would arrive back at $P_{\text{best}}$ exactly at the last round while having maximized her exploration range.

Looking into the behavioural data, however, it seems that participants only use a certain fraction of the optimal exploration range (see, e.g. the example in figure 5b). That is, they start returning to their best option too early with respect to $D_{\text{optimal}}$. Some participants even stop their exploration immediately when a new $P_{\text{best}}$ is discovered, without using any of the remaining rounds for further exploration. In contrast, it also happens that participants continue searching longer than $D_{\text{optimal}}$ and hence do not return to $P_{\text{best}}$ at all.

To study this process, we measured the safety level $S$ of our participants, defined as the fraction of the optimal exploration range they used before returning to $P_{\text{best}}$. That is, $S = R/D_{\text{optimal}}$, where $R$ is the distance a participant has actually moved away from $P_{\text{best}}$ before returning to it. With this definition, $S = 1$ indicates that all remaining rounds were used efficiently, $S = 0$ indicates that the exploration has stopped immediately after the discovery of the peak and $S > 1$ indicates that the participant did not return to $P_{\text{best}}$ at all. Figure 6 shows the values of the safety levels $S$ observed in our experiment, as a function of the pay-off value $X_{\text{best}}$.

Figure 6a reveals multiple zones of interest: (i) on the upper part of the figure, the stronger density of data points around $S = 0.9$ indicating that participants often continued their search up to about 90% of the optimal exploration range; (ii) on the lower part of the figure, however, the stronger density of data points along $S = 0$ indicates that participants often decided to stop their search immediately after discovering a new peak. In this case, the probability to stop seems to be linearly increasing with the pay-off value $X_{\text{best}}$. The case where $S > 1$ is most frequent when the value of $X_{\text{best}}$ is low (mean pay-off for $S > 1 = 9.38$, s.d. = 9.05). In sum, participants tend to stop immediately if a sufficiently good peak is found and otherwise continue their exploration before coming back to it with a certain safety time. If the pay-off of the discovered peak is too low, however, they do not return to it.

Formally, the exploration range $R$ around a newly discovered best solution $P_{\text{best}}$ can therefore be defined as $R = 1$ with a probability of $k \times X_{\text{best}}$, $R = \infty$ with a probability of $1 - l \times X_{\text{best}}$ and $R = S_0 \times$

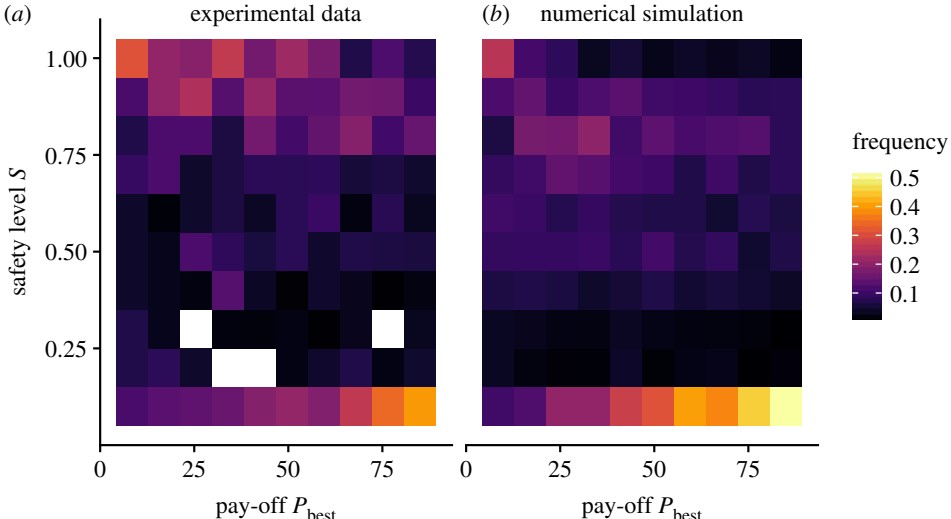

**Figure 6.** Safety level $S$ as a function of the pay-off $X_{best}$, (a) as observed in the experimental data and (b) obtained from numerical simulations. All incidents where $D_{optimal} \leq 2$ or $S > 1$ were excluded, as these cases neither show return nor staying behaviour. In total, the removed incidents account for 26% of the data, of which 75% are cases where no pay-off bigger than zero is found at all. To generate the heatmap, we first determined the safety level $S$ for all trajectories where the participants returned to a previously visited $P_{best}$. We then calculated the relative frequency for $S$ given a certain $X_{best}$, which are indicated by the colour coding.

$D_{optimal}$ otherwise, where $D_{optimal} = \lfloor T_{left}/2 \rfloor$ as previously defined. The parameter $k$ describes the linear influence of $X_{best}$ on the stopping probability (fitted value $k = 0.0051$, s.e. = 0.0003 using a linear model applied to cases where $S = 0$). The parameter $l$ describes the linear influence of $X_{best}$ on the probability to ignore a peak (fitted value $l = -0.027$, s.e. = 0.005 using a linear model applied to cases where $S > 1$). The parameter $S_0$ is the safety level that a person adopts and is sampled from a truncated normal distribution $\mathcal{N}(\mu, \sigma^2)$ bounded between zero and one (fitted values $\mu = 0.806$, s.e. = 0.04 and $\sigma^2 = 0.364$, s.e. = 0.03 using a maximum-likelihood estimator applied to all observations with $S > 0$ and $S \leq 1$).

How can the exploration model described in the previous section be extended to account for this stopping process? Remarkably, only one additional cue in the TTB heuristic is sufficient to do so: the *exploration-radius cue*, indicating whether a given option lies within the exploration radius $R$ or not. The general model is therefore composed of four cues—exploration-radius, not-visited, pay-off and novelty—that are considered one at a time and in this order. In other words, the model simply ensures that only options that are not too far from the current best pay-off are considered. Interestingly, the returning behaviour does not need to be explicitly implemented in the model. Instead, it emerges naturally. In fact, the exploration range $R$ shrinks towards the current best solution $P_{best}$ as the end of the allocated time is approaching (due to the dependency of $R$ on the remaining time $T_{left}$). Therefore, after a certain time, only decisions towards $P_{best}$ are considered, gradually driving the individual back to its best solution.

The model reproduces the trends observed in the experimental data, in terms of pay-offs and individual behaviours (figure 7) and shows a similar relationship between the pay-off $X_{best}$ and the safety level $S$ (figure 6).

## 3.3. Combined phase

In the third experimental phase, participants performed the search task in two-dimensional landscapes (as in the exploration phase) but were rewarded based on the pay-off of the field they occupy at the last round (as in the exploitation phase). This phase, therefore, combines the exploration and the exploitation processes that we previously studied separately, and allows us to evaluate the full model that we have elaborated.

The average pay-off in the rich landscapes is higher than in the poor landscapes (average final pay-off 0.36, with an s.d. = 0.20 and 0.24 with an s.d. = 0.29 in the rich and poor landscapes, respectively). When looking at the density maps, the X-shaped exploration pattern is visible in the poor landscapes but not in the rich landscape, similar to the exploration phase (figure 8).

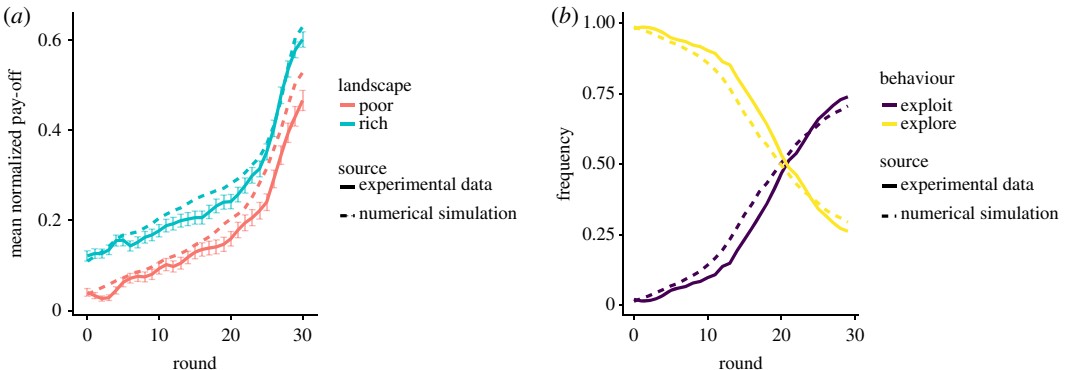

**Figure 7.** Performance and behaviour in the exploitation phase. (*a*) Observed and simulated average normalized pay-off as a function of time. (*b*) Frequency of exploration (moving away from $P_{best}$) and exploitation (moving towards or staying at $P_{best}$) as a function of time. Error bars for the experimental data indicate the standard error of the mean.

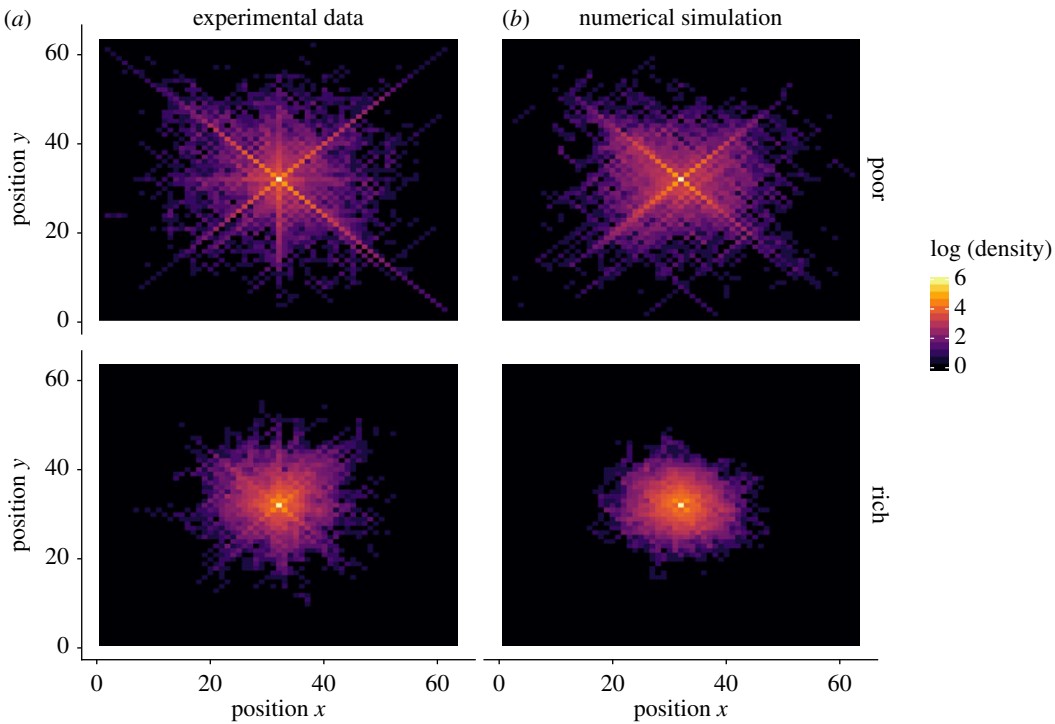

**Figure 8.** Density maps for poor (upper line) and rich (bottom line) landscapes in the combined phase, as observed in the experimental data (*a*) and obtained from numerical simulations (*b*). The colour coding indicates how often a given position ($x$, $y$) has been visited at the aggregate level, represented in the logarithmic scale. The starting position is located in the middle of the map, at coordinates (32, 32). For the simulations, we randomly selected the same number of trajectories as in the behavioural data to ensure comparable density scales.

Does our full model reproduce these patterns? The same fitting procedure as in the exploitation phase yields the parameter values $k = 0.011$ (s.e. = 0.003), $l = -0.022$ (s.e. = 0.012), $\mu = 0.386$ (s.e. = 0.02) and $\sigma^2 = 0.275$ (s.e. = 0.01). Interestingly, these values are different from those fitted in the exploitation phase. The decrease of $k$ (the influence of $X_{best}$ on the stopping probability) and $\mu$ (the mean safety level) reflect the fact that participants were satisfied with a lower pay-off and adopted a lower safety level than in the exploitation phase. This can be explained by the greater complexity of the task, which reduced the participants' willingness to move away from a discovered solution [2]. Despite this overall decrease of $\mu$, we find a strong correlation between the participants' safety level in this phase and in the exploitation phase (Pearson's correlation = 0.42, d.f. = 47, $p < 0.003$, see also electronic supplementary

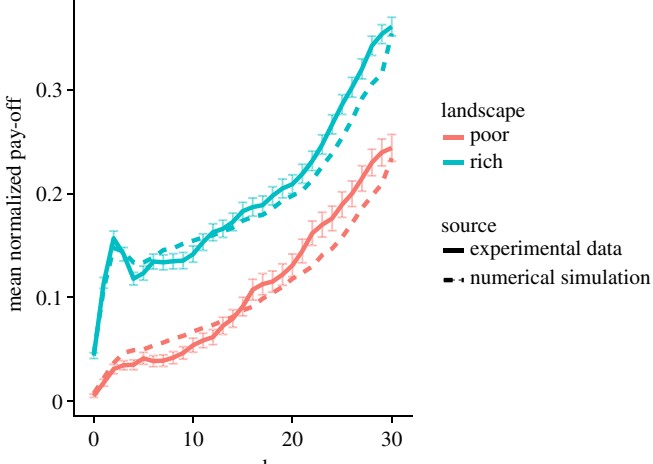

**Figure 9.** Performances in the combined phase. Evolution of the average normalized pay-off, aggregated over all participants, as a function of time in the experimental data and model simulations. Error bars for the experimental data indicate the standard error of the mean.

material, figure A6), suggesting that the difference between the individual safety levels remained somewhat stable.

The model predicts the emerging X-shaped exploration pattern (figure 8) as well as the evolution of the average pay-offs (figure 9). Overall, this confirms that the full model has captured some of the key mechanisms involved in the search process of the participants.

## 3.4. Model comparison

How well does our full model perform compared to alternative modelling approaches? Because most models in the literature either focus on the exploration (e.g. [9,19,31]) or the exploitation mechanism (e.g. [10,11,20]), we tested these two main components separately. For this, we first kept the exploitation mechanism unchanged, and tested five different models for the exploration phase: (i) our *take-the-best* model, (ii) a *probabilistic* variation of that model in which the individual chooses to rely on the pay-off cue or the visibility cue based on the most-rewarding neighbouring solution [32,33], (iii) a typical *hill-climbing* model where exploration is always directed towards the most-rewarding adjacent position [34], (iv) a *blind search* model in which the exploration is only guided by novelty, and (v) a *random search* model in which the next position is randomly chosen among the adjacent solutions. For the probabilistic model, we fit a logistic regression to predict the probability of using the pay-off cue over the novelty cue, depending on the most-rewarding neighbouring solution. The probability to rely on the pay-off cue is $1 - 1/e^{-1.509+0.301pn_{max}}$, where $pn_{max}$ is the pay-off of the most-rewarding neighbouring solution (see electronic supplementary material, figure A7).

The results are presented in table 3. It appears that take-the-best predicts the experimental data best, in terms of both the pay-off and the density map. Specifically, the three models that use the pay-off cue (take-the-best, probabilistic, and hill-climbing) are in good agreement with the observed pay-offs pattern. However, the X-shaped pattern observed in the experimental density maps can only be predicted when using the novelty cue (see also the electronic supplementary material, figures A2 and A3).

We then evaluated the exploitation mechanism analogously, by keeping the exploration mechanism unchanged, and tested four possible models for the exploitation phase: (i) a *normative* model in which the individuals return to their best solution with no safety time, (ii) an *early-stop* model in which the individuals stop exploring whenever the first peak is found [28], (iii) our *returning model*, and (iv) *simple returning*, a variation of our model where individuals do not ignore peaks with low pay-off [35]. The results are shown in table 4. The normative model strongly deviates from the observed pay-off pattern, as participants tend to stop exploration and return to their best solution much earlier than the simulated agents. Likewise, the early-stop model produces pay-off patterns that are inconsistent with the observations (see electronic supplementary material, figures A4 and A5). The two remaining models have similar performances: the returning model predicts the experimental data slightly better, whereas the simple returning model has one less free parameter [36]. Remaining deviations from

**Table 3.** Comparison of different exploration mechanisms. The first column indicates how much the model predictions deviate from the observations with regard to the pay-off pattern shown in figure 9. Formally, $dist_{pay-off}$ measures the squared difference between the observed and the predicted pay-off curves. The second column indicates the deviation of the predictions with regard to the density map (figure 8). Here, $dist_{density}$ measures the absolute average difference between each position of the observed and the predicted density maps. To assess the *prediction accuracy* we used a $k$-fold cross-validation ($k = 5$) and out-of-sample predictions for calculating $dist_{pay-off}$ and $dist_{density}$ [36]. The last two columns indicate the number of free parameters of the model and the number of cues used in the exploration mechanism. All five exploration models are tested in combination with the *returning model* for exploitation (table 4).

| name | $dist_{pay-off}$ | $dist_{density}$ | parameters | cues |
|------|------|------|------|------|
| take-the-best | 2.83 | 0.68 | 1 | 3 |
| probabilistic | 3.34 | 0.77 | 2 | 3 |
| hill-climbing | 5.07 | 0.79 | 0 | 2 |
| blind search | 69.6 | 1.06 | 0 | 2 |
| random search | 82.0 | 0.93 | 0 | 0 |

**Table 4.** Comparison of different exploitation mechanisms. The values of $dist_{pay-off}$ and $dist_{density}$ measure the difference between observations and model's predictions, in terms of pay-off patterns and density maps, respectively (see table 3 for formal definitions). All four exploitation models are tested in combination with the *take-the-best model* for exploration (table 3).

| name | $dist_{pay-off}$ | $dist_{density}$ | parameters | cues |
|------|------|------|------|------|
| returning | 2.83 | 0.68 | 4 | 1 |
| simple returning | 3.48 | 0.73 | 3 | 1 |
| early-stop | 17.7 | 0.75 | 0 | 0 |
| normative | 39.4 | 0.85 | 0 | 1 |

observed data in the TTB model can be attributed to some of the model's simplifications, such as ignoring a possible time-dependency of the safety level $S$ (see electronic supplementary material, figure A8).

# 4. Discussion

Search problems can vary on many different dimensions. We investigated how people search for a rewarding outcome in problems characterized by gradual exploration (i.e. when jumps between distant solutions are not allowed) and a temporal separation between exploration and exploitation. This type of search is relevant for numerous real-life situations, such as visual search, spatial search and most problem-solving situations [5,14,24,25]. On that account, we have developed a dedicated experimental design enabling us to isolate the exploration from the exploitation mechanisms. We then modelled these two components separately, before merging them in a full and comprehensive model of search.

We described the participants' behaviour by means of the TTB heuristic. With that approach, four cues can describe the exploration and exploitation behaviour of the participants: the exploration-radius cue, the non-visited cue, the pay-off cue and the novelty cue. While TTB constitutes a valid model to describe how people make decisions between two options [37], we have shown that it can also be used to describe search behaviours.

Recent research on human search behaviours distinguishes between directed and undirected exploration [19,31,32]. In multi-armed bandit tasks undirected exploration refers to the stochasticity of the search process causing random exploration decisions. By contrast, directed exploration seeks out solutions that are informative about the underlying reward distribution [32]. Both components play an important role in solving the exploration–exploitation dilemma and are often considered as 'two core components of exploration' [31]. Our full model stands along the same lines: the noise parameter $\epsilon$ accounts for undirected exploration, whereas the novelty cue guides the exploration towards unknown regions of the solution space and hence accounts for directed exploration.

Our experimental data revealed that the decision to stop exploration and start exploiting a solution relies on a satisficing behaviour [35]. That is, participants exhibited a tendency to terminate their exploration immediately when a good enough solution is found, even though more time was still available. In this context, the idea of satisficing seems inefficient as individuals overlooked an opportunity to explore new solutions and come back to their best solution only when time was running out. Research has shown that satisficing—and thus deliberately reducing the exploration range—can be adaptive when the number of solutions is much larger than the individual's search horizon [35] and when exploration is too difficult compared to its expected reward gain [38]. In our design, continuing the exploration after the discovery of a sufficiently good solution creates a risk of not finding it again when returning to it. This uncertainty should grow as the size of the solution space increases. In agreement with this idea, we observed that participants were satisfied earlier in two-dimensional landscapes than in uni-dimensional ones.

We also observed a correlation between the participants' safety levels across the different phases of the experiment. That is, participants who exhibited a higher safety level in the exploitation phase were also more likely to show a higher safety level in the combined phase (and respectively, for a lower safety level). Interestingly, the safety level reflects the participant's propensity to take risks: the longer individuals continue to explore after finding a peak, the higher their risk of not finding the peak again. The consistency of the observed safety levels across phases thus agrees with risk research showing that people's risk preferences tend to be stable over time and tasks [39].

In our experiments and simulations, we systematically compared two specific types of search environments: rich and poor landscapes, which differ in the number of peaks that are present. Nevertheless, other structural aspects of the search environment could be varied as well, such as the peaks widths, heights or locations. This last feature is particularly useful to create 'patchy' landscapes in which peaks tend to be clustered in specific regions of the search space [40]. Additional simulations presented in figures A2–A4 in the electronic supplementary materials show that our heuristic model seems to behave realistically in this new type of environments, although these predictions still need to be tested experimentally. Future work will investigate if heuristic models similar to our TTB approach could be generalized to describe people's search behaviour in different types of environments, including such patchy landscapes.

Ethics. Participants gave informed consent to the experiment. The experimental procedure was approved by the Ethics Committee of the Max Planck Institute for Human Development.

Data accessibility. Anonymous participant data, model simulation and experimental code are available at https://github.com/cuehs/search_heuristic and are archived within the Zenodo repository https://doi.org/10.5281/zenodo.3331373 [41].

Authors' contributions. K.S.Y. and M.M. designed research, performed research, analysed data and wrote the manuscript.

Competing interests. We declare we have no competing interests.

Funding. This research was supported by a grant from the German Research Foundation as part of the priority program on 'New Frameworks of Rationality' (SPP 1516) awarded to Ralph Hertwig (grant no. HE 2768/7-2). The funders had no role in study design, data-collection and analysis, decision to publish or preparation of the manuscript.

Acknowledgements. We thank the Adaptive Rationality research group and the three anonymous reviewers for the valuable feedback.

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
