## [Reviewer comments · Royal Society Open Science]

Review History

RSOS-190529.R0 (Original submission)

Review form: Reviewer 1

Is the manuscript scientifically sound in its present form?

Yes

Are the interpretations and conclusions justified by the results?

Yes

Is the language acceptable?

Yes

Is it clear how to access all supporting data?

Yes

Do you have any ethical concerns with this paper?

No

Have you any concerns about statistical analyses in this paper?

No

Recommendation?

Accept with minor revision (please list in comments)

Comments to the Author(s)

This is a strong paper on a very interesting decision-making topic. I carefully read through the manuscript and have no major concerns. This is a well-done piece with an up-to-date reference list. The manuscript has a clear structure and uses concise language. I only have a few points that I would like to see addressed:

- 1) The manuscript would benefit from more clearly fleshing out what specific (real-world) choice problems come with 'gradual exploration' and a 'temporal separation between exploration and exploitation'. Adding this information beyond merely stating that these choice situations occur in visual search, spatial search and problem-solving situations would make the manuscript more appealing to a broader audience.
- 2) Please add more information regarding the instructions that participants received and what training they completed. What was the "cover story" for the search problem that participants completed?
- 3) Many other choice problems that the authors describe in their paper are imbedded in very specific choice environments. Often, the statistical regularities of these choice environments tell us a lot about what participants should or should not do. While the choice environment is somewhat specified in the current study, I could not fully make sense of what drove the distribution of normalized payoffs that was used. Is there a particular reason that a more or less evenly dispersed distribution was chosen for the poor and rich landscape? Would the obtained results be the same if these payoff distributions were to be changed – say into a random distribution of payoff distributions with clusters/patches of resources?

Review form: Reviewer 2

Is the manuscript scientifically sound in its present form?

No

Are the interpretations and conclusions justified by the results?

No

Is the language acceptable?

Yes

Is it clear how to access all supporting data?

Not Applicable

Do you have any ethical concerns with this paper?

No

Have you any concerns about statistical analyses in this paper?

No

Recommendation?

Major revision is needed (please make suggestions in comments)

Comments to the Author(s)

This work investigates search on a two-dimensional array and attempts to develop a search heuristic that describes the underlying search process. Three experimental settings are used, involving an exploration, exploitation, and combined phase. The authors suggest that the proposed heuristic (TTB) can be used to describe search behaviors.

This work takes a novel approach to understanding search behavior. However, I think it is premature for several reasons as I outline below:

1. Search paradigm and environment structure: The environments explored and the paradigm used to study search represent a small subset of what many people might call search. Both the poor and rich environment in Figure 1 are fairly sparse and clumps in these environments seem to rarely represent more than 9 pixels. This means it will rarely be meaningful to return to a previously encountered cluster, and clusters can be harvested fairly straightforwardly upon encounter. The TTB heuristic captures this harvesting phase and the movement away from previously visited solutions. But moving away from previously visited cells (a combination of the non-visited and novelty cue) and harvesting those you encounter is not what I would call search in any but the most trivial sense. This would not capture one of the most commonly observed search patterns area-concentrated search, because it would not move back to previously visited clusters after it left them. The TTB search process has no memory, much like a Lévy flight, which throws this heuristic into the cognitive capacities of a bacterium. BUT, a bacterium has a short memory, and will turn around if it starts to move down a resource gradient. TTB has no such memory. However, the search environment investigated doesn't have clusters sufficiently large to engage patch or cluster based strategies, making them perhaps unrealistic even for bacteria. I would suggest including a search environment with larger clusters and without the ability to see neighboring cells. The problem is that with this latter constraint (the inability to see neighboring cues), the TTB heuristic is not realistic. This tells us something about the TTB heuristic. It is not a search strategy, but a harvest strategy. TTB can only take what it has already found. The question of search is really 'how does one look?'
2. Why not compete heuristics? The heuristic described is straightforward. But is it the only possibility? I think the work would be improved by competing heuristics against one another. This will also tease apart the differences between different environments. For example, consider a 'harvest' and 'explore' strategy. This strategy takes the best, after if it finds it, same as the payoff cue, when it is visible in the surrounding cells. Otherwise, it always moves away from the last visited-cue. One could add that it moves diagonally away from the last visited cue. This last constraint though is an artefact of having space distributed as a pixel array. Alternatively, one could try another heuristic, if one finds oneself in a field of view without resources immediately after being in a field of view with resources, then turn around. This is the bacterium's strategy and it makes sense in a clumpy world (see for example Pacheco-Cobos et al. 2019 and Hutchinson et al 2012 cited therein). One could also develop some Levy walk like heuristics.
3. I think the analysis is good and I appreciate the details. I would like to know how often the heuristic (and its competitors) make a choice consistent with the observed behaviour of the participants.
4. Is $\epsilon = 0.17$ true for all environments?

Review form: Reviewer 3 (Mehdi Khamassi)

Is the manuscript scientifically sound in its present form?

No

Are the interpretations and conclusions justified by the results?

No

Is the language acceptable?

Yes

Is it clear how to access all supporting data?

Yes

Do you have any ethical concerns with this paper?

No

Have you any concerns about statistical analyses in this paper?

Yes

Recommendation?

Major revision is needed (please make suggestions in comments)

Comments to the Author(s)

The manuscript entitled 'Search as a simple take-the-best heuristic' by Yahosseini and Moussaïd presents a navigation experiment within a grid-world ('landscape') where subjects are instructed that a single position corresponds to the best solution (most rewarding one). The landscape environments are structured in that a local gradient was imposed around each peak of reward. The experiment comprises three phases aimed at investigating how human subjects deal with exploration/search strategies (1) when only gradual movements are possible (i.e. jumps are not feasible), (2) when they decide to terminate the exploration and start exploiting one solution when these two phases are temporally separated, and (3) when confronted to both rich and poor solution spaces. Every phase consisted of 20 landscapes (divided in 10 rich and 10 poor landscapes) played for 30 rounds each. Results show that these two behavioural components are well described by a simple bounded rationality model based on the take-the-best heuristic (Gigerenzer & Goldstein 1996, 1999). The model is composed of three hierarchical heuristics (not-visited cue, payoff cue, novelty cue) when applied to the exploration phase of the experiment, and later extended to include another heuristic (exploration-radius cue) to account for the exploitation phase.

Overall, the experimental results are interesting and clear. Nevertheless, some statistical tests are missing here and there (see detailed comments below). Moreover, the computational model is a bit unsatisfying so far in that: (1) it looks a bit adhoc, by incorporating heuristics that are specific to the present data. (2) the model is not constant between phases, but extended phase after phase to account for all the data. It would have been more compelling to present a single model with a general principle/mechanism which can account for the experimental data in all phases of the experiment. Moreover, we don't really learn much more than the fact that human subjects adapt their strategy depending on the specific instructions of each phase of the task. But aren't there instead some general tendencies of human search behavior which could be highlighted here and which would have been overlooked by the existing literature? After reading the paper, I am not fully convinced. Nevertheless, I suggest below some directions along which I think the paper could be satisfyingly improved. Finally, two other drawbacks with the current version: (3) the model simulations sometimes differ from the subjects' behavior observed in the experiments. So the model is not completely satisfying yet. These differences should be quantified, discussed, and if possible overcome. (4) the model is not compared to alternative models from the literature. This paper could bring a stronger contribution to the field if the authors could compare their model with alternative models from the literature. I have cited a few examples in my detailed comments below. I am sure the authors can find additional central models from their own field of research

and compare all these models with quantitative methods. I also cite below a methodological paper by Nathaniel Daw about current state-of-the-art model comparison methods. This way, the paper could show that model X cannot account for the data while model Y can. This would be useful in that the novel experimental data presented here could contribute to the refutation of existing models/theories. Without this, the model for the moment sounds a bit like an adhoc solution to account for the specific data gathered in this experiment.

Major points

Could the authors present some statistical test to assess the significance of the observation that ‘In both environments, participants avoided revisiting a previously visited solutions (on average 14% of the solutions were visited more than once).’ (page 5, lines 56-58)? In particular, is this percentage significantly higher than what can be obtained by chance (for instance with a random walk)?

In Figure 2, rich landscape, experimental data, it seems that the X-shape is present but only within a small region around the center. In contrast, the model does not seem to present this feature. Is there a way the quantitatively assess the presence of the X-shape and then evaluate whether the model can produce to some extent also in the rich landscape?

In Figure 3C, for poor landscapes, the experimental data and the model seem to produce qualitatively different patterns. Can the authors quantitatively compare them and discuss this difference?

The presented model is composed of one particular strategy, which arbitrarily ranks payoffs above novelty in terms of priority. However, it would be interesting to study (1) whether other models like UCB (Auer et al 2002) or uncertainty bonus models (Daw O’Doherty et al 2006 Nature; Frank et al 2009 Nature Neuroscience; Wilson et al 2014 already cited; Gershman 2018 already cited; Cogliati-Dezza et al 2017 Scientific Reports) or similar models, which decide based on a linear combination of payoff and ‘novelty’ (or rather ‘uncertainty’ in their case, which is related in the present task); and (2) individual differences between subjects by testing whether different models (implementing different search strategies) are best suited to fit different subjects’ data. As the authors mention in the discussion, their model roughly stands along the same line as models distinguishing directed and random exploration, because their model has a noise parameter and a novelty cue element. Nevertheless, the model arbitrarily sets a hierarchy between these two components while the above mentioned models use a weighting parameter between the two components, this parameter being a free parameter optimized per subject: different subjects give a different weight to novelty/uncertainty than others.

The model has 0 chance of revisiting a previously visited state, while this happens sometimes in subjects (14% of the time). Why not adding another noise parameter to allow for this? Would this second model perform better than the present one? During such a model comparison process, the authors should consider a way to penalize for model complexity, that is, the number of free parameters (Daw, N. D. (2011). Trial-by-trial data analysis using computational models. *Decision making, affect, and learning: Attention and performance XXIII*, 23(1).).

Figure 5B shows an example trajectory by a subject who could have explored a bit further (i.e., 5 steps more along the x axis) and would still have been able to get back to the optimum before the end of the experiment. As the authors note, this is an example of a suboptimal strategy. First, could the authors analyze the proportion of subjects that had a close to optimal strategy in this respect, and those who hadn’t? In particular, I am wondering if the tendency to perform a suboptimal strategy with this respect is something stable within a given subject, or whether

subjects sometimes performed an optimal one, and sometimes a suboptimal one. Second, it seems to that the risk to explore further after discovering an optimum (X_{best}) depends on the remaining number of rounds (T_{left}). Thus, maybe the analysis of the subjects' tendency (Figure 6) would be different if the authors could split the data into early trials ($T_{\text{left}} \geq 15$ to 20) and late trials ($T_{\text{left}} < 15$ to 20), or something like that. Figure 7B illustrates something similar to this intuition. By the way, the authors should specify what they mean by « exploration » and « exploitation » in Figure 7, which I think is simply moving away from X_{best} versus moving towards it. This should be clearly indicated to the reader. Anyway, What I mean is slightly different and is a prediction that the subjects should have the tendency to less perform the optimal strategy (thus explore) when $T_{\text{left}} \geq 15$, even when they still have time to do it and it would be optimal to do so, than when $T_{\text{left}} < 15$. It would be interesting if the authors could dig a bit more in this direction.

To me, Figure 6 shows a striking difference between experimental data and model for small payoffs values (upper-left corner): the experimental data show that the subjects tend to frequently display high safety values (close to 1) in this area, while the model doesn't. Could the authors discuss this difference, try to explain it, and think of an alternative model which could account for this tendency?

I disagree with the authors that the model well captures the evolution of the average payoffs (figure 9) in the combined phase. One can see large series of rounds (especially between round 4 and round 15) where the model seems to perform significantly differently than the subjects. The authors should first perform some statistical tests to assess when these differences are significantly different. Moreover, the authors should be clearer here whether they used the version of the model with only 3 cues (from the exploration phase) or the one with 4 cues (from the exploitation phase). In addition, it is not satisfying to change the model between phases. A single model should explain behaviors in all phases. Could the authors test the model with 4 cues on the data of the exploration phase and discuss the results? Last but not least, this work would bring a stronger contribution to the field if the authors could compare their model with alternative models from the literature. I have cited a few examples above. I am sure the authors can find additional central models from their own field of research and compare all these models with quantitative methods. This way, the paper could show that model X cannot account for the data while model Y can. This would be useful in that the novel experimental data presented here could contribute to the refutation of existing models/theories. Without this, the model for the moment sounds a bit like an adhoc solution to account for the specific data gathered in this experiment.

The discussion section is bit short and does not bring enough perspective with respect to model predictions, what we learned from this study which was completely new compared to previous studies, what we learn about computational mechanisms used in existing models, etc. If the paper could include some model comparison, then the discussion could address the results of this model comparison and thus bring a wider perspective on the results.

Minor points

For clarity, in the legend of Figure 1, the authors should specify that the two example trajectories were obtained during the exploration phase.

Page 7, line 41, at this stage of reading, it is not clear what the authors mean by 'ranking cues according to their validity'. Could they please be more precise here?

Page 10, lines 49-50, the author should cite Figure 8 at the end of the following sentence: 'When looking at the density maps (figure X), the X-shaped exploration pattern is visible in the poor landscapes but not in the rich landscape, similar to the exploration phase.'

In terms of state of the art, I think that the tourist problem, with sharp transition between exploration and exploitation, as it is stated to present the contribution of the paper, is tightly related to foraging problems, which are studied by a literature which is overlooked here. The authors could benefit from looking at some of the related literature, both on the purely behavioural side, as well as on the behavioural neuroscience side: D. W. Stephens, J. R. Krebs, Foraging Theory (Princeton Univ. Press, Princeton, NJ, 1986). Hayden, B. Y., Pearson, J. M., & Platt, M. L. (2011). Neuronal basis of sequential foraging decisions in a patchy environment. *Nature neuroscience*, 14(7), 933. Kolling, N., Behrens, T. E., Mars, R. B., & Rushworth, M. F. (2012). Neural mechanisms of foraging. *Science*, 336(6077), 95-98.

Typos

Page 2, line 30: 'the separation of exploration and exploration' -> exploration and exploitation.

Page 6, line 49, 'an efficient strategy to extent one's exploration range' -> .. to extend ..

Decision letter (RSOS-190529.R0)

18-Jun-2019

Dear Mr Yahosseini,

The editors assigned to your paper ("Search as a simple take-the-best heuristic") have now received comments from reviewers. We would like you to revise your paper in accordance with the referee and Associate Editor suggestions which can be found below (not including confidential reports to the Editor). Please note this decision does not guarantee eventual acceptance.

Please submit a copy of your revised paper before 11-Jul-2019. Please note that the revision deadline will expire at 00.00am on this date. If we do not hear from you within this time then it will be assumed that the paper has been withdrawn. In exceptional circumstances, extensions may be possible if agreed with the Editorial Office in advance. We do not allow multiple rounds of revision so we urge you to make every effort to fully address all of the comments at this stage. If deemed necessary by the Editors, your manuscript will be sent back to one or more of the original reviewers for assessment. If the original reviewers are not available, we may invite new reviewers.

When submitting your revised manuscript, you must respond to the comments made by the referees and upload a file "Response to Referees" in "Section 6 - File Upload". Please use this to

document how you have responded to the comments, and the adjustments you have made. In order to expedite the processing of the revised manuscript, please be as specific as possible in your response.

- Data accessibility

If you wish to submit your supporting data or code to Dryad (<http://datadryad.org/>), or modify your current submission to dryad, please use the following link:
<http://datadryad.org/submit?journalID=RSOS&manu=RSOS-190529>

- Competing interests

- Authors' contributions

- Acknowledgements

- Funding statement

Kind regards,

Alice Power

Editorial Coordinator

on behalf of Dr Narayanan Srinivasan (Associate Editor) and Essi Viding (Subject Editor)

Associate Editor's comments (Dr Narayanan Srinivasan):

Three experts have commented on the paper. The authors are requested to address all the concerns of reviewers point by point.

Comments to Author:

Reviewers' Comments to Author:

Reviewer: 1

Comments to the Author(s)

This is a strong paper on a very interesting decision-making topic. I carefully read through the manuscript and have no major concerns. This is a well-done piece with an up-to-date reference list. The manuscript has a clear structure and uses concise language. I only have a few points that I would like to see addressed:

- 1) The manuscript would benefit from more clearly fleshing out what specific (real-world) choice problems come with 'gradual exploration' and a 'temporal separation between exploration and exploitation'. Adding this information beyond merely stating that these choice situations occur in visual search, spatial search and problem-solving situations would make the manuscript more appealing to a broader audience.
- 2) Please add more information regarding the instructions that participants received and what training they completed. What was the "cover story" for the search problem that participants completed?
- 3) Many other choice problems that the authors describe in their paper are imbedded in very specific choice environments. Often, the statistical regularities of these choice environments tell us a lot about what participants should or should not do. While the choice environment is somewhat specified in the current study, I could not fully make sense of what drove the distribution of normalized payoffs that was used. Is there a particular reason that a more or less evenly dispersed distribution was chosen for the poor and rich landscape? Would the obtained results be the same if these payoff distributions were to be changed – say into a random distribution of payoff distributions with clusters/patches of resources?

Reviewer: 2

Comments to the Author(s)

This work investigates search on a two-dimensional array and attempts to develop a search heuristic that describes the underlying search process. Three experimental settings are used, involving an exploration, exploitation, and combined phase. The authors suggest that the proposed heuristic (TTB) can be used to describe search behaviors.

This work takes a novel approach to understanding search behavior. However, I think it is premature for several reasons as I outline below:

1. Search paradigm and environment structure: The environments explored and the paradigm used to study search represent a small subset of what many people might call search. Both the poor and rich environment in Figure 1 are fairly sparse and clumps in these environments seem to rarely represent more than 9 pixels. This means it will rarely be meaningful to return to a previously encountered cluster, and clusters can be harvested fairly straightforwardly upon encounter. The TTB heuristic captures this harvesting phase and the movement away from previously visited solutions. But moving away from previously visited cells (a combination of the non-visited and novelty cue) and harvesting those you encounter is not what I would call search in any but the most trivial sense. This would not capture one of the most commonly observed search patterns area-concentrated search, because it would not move back to previously visited clusters after it left them. The TTB search process has no memory, much like a Lévy flight, which throws this heuristic into the cognitive capacities of a bacterium. BUT, a bacterium has a short memory, and will turn around if it starts to move down a resource gradient. TTB has no such memory. However, the search environment investigated doesn't have clusters sufficiently large to engage patch or cluster based strategies, making them perhaps unrealistic even for bacteria. I would suggest including a search environment with larger clusters and without the ability to see neighboring cells. The problem is that with this latter constraint (the inability to see neighboring cues), the TTB heuristic is not realistic. This tells us something about the TTB heuristic. It is not a search strategy, but a harvest strategy. TTB can only take what it has already found. The question of search is really 'how does one look?'
2. Why not compete heuristics? The heuristic described is straightforward. But is it the only possibility? I think the work would be improved by competing heuristics against one another. This will also tease apart the differences between different environments. For example, consider a 'harvest' and 'explore' strategy. This strategy takes the best, after if it finds it, same as the payoff cue, when it is visible in the surrounding cells. Otherwise, it always moves away from the last visited-cue. One could add that it moves diagonally away from the last visited cue. This last constraint though is an artefact of having space distributed as a pixel array. Alternatively, one could try another heuristic, if one finds oneself in a field of view without resources immediately after being in a field of view with resources, then turn around. This is the bacterium's strategy and it makes sense in a clumpy world (see for example Pacheco-Cobos et al. 2019 and Hutchinson et al 2012 cited therein). One could also develop some Levy walk like heuristics.
3. I think the analysis is good and I appreciate the details. I would like to know how often the heuristic (and its competitors) make a choice consistent with the observed behaviour of the participants.
4. Is $\epsilon = 0.17$ true for all environments?

Reviewer: 3

Comments to the Author(s)

The manuscript entitled 'Search as a simple take-the-best heuristic' by Yahosseini and Moussaïd presents a navigation experiment within a grid-world ('landscape') where subjects are instructed that a single position corresponds to the best solution (most rewarding one). The landscape environments are structured in that a local gradient was imposed around each peak of reward. The experiment comprises three phases aimed at investigating how human subjects deal with exploration/search strategies (1) when only gradual movements are possible (i.e. jumps are not feasible), (2) when they decide to terminate the exploration and start exploiting one solution when these two phases are temporally separated, and (3) when confronted to both rich and poor solution spaces. Every phase consisted of 20 landscapes (divided in 10 rich and 10 poor landscapes) played for 30 rounds each. Results show that these two behavioural components are well described by a simple bounded rationality model based on the take-the-best heuristic (Gigerenzer & Goldstein 1996, 1999). The model is composed of three hierarchical heuristics (not-visited cue, payoff cue, novelty cue) when applied to the exploration phase of the experiment, and later extended to include another heuristic (exploration-radius cue) to account for the exploitation phase.

Overall, the experimental results are interesting and clear. Nevertheless, some statistical tests are missing here and there (see detailed comments below). Moreover, the computational model is a bit unsatisfying so far in that: (1) it looks a bit adhoc, by incorporating heuristics that are specific to the present data. (2) the model is not constant between phases, but extended phase after phase to account for all the data. It would have been more compelling to present a single model with a general principle/mechanism which can account for the experimental data in all phases of the experiment. Moreover, we don't really learn much more than the fact that human subjects adapt their strategy depending on the specific instructions of each phase of the task. But aren't there instead some general tendencies of human search behavior which could be highlighted here and which would have been overlooked by the existing literature? After reading the paper, I am not fully convinced. Nevertheless, I suggest below some directions along which I think the paper could be satisfyingly improved. Finally, two other drawbacks with the current version: (3) the model simulations sometimes differ from the subjects' behavior observed in the experiments. So the model is not completely satisfying yet. These differences should be quantified, discussed, and if possible overcome. (4) the model is not compared to alternative models from the literature. This paper could bring a stronger contribution to the field if the authors could compare their model with alternative models from the literature. I have cited a few examples in my detailed comments below. I am sure the authors can find additional central models from their own field of research and compare all these models with quantitative methods. I also cite below a methodological paper by Nathaniel Daw about current state-of-the-art model comparison methods. This way, the paper could show that model X cannot account for the data while model Y can. This would be useful in that the novel experimental data presented here could contribute to the refutation of existing models/theories. Without this, the model for the moment sounds a bit like an adhoc solution to account for the specific data gathered in this experiment.

Major points

Could the authors present some statistical test to assess the significance of the observation that 'In both environments, participants avoided revisiting a previously visited solutions (on average 14% of the solutions were visited more than once).' (page 5, lines 56-58)? In particular, is this percentage significantly higher than what can be obtained by chance (for instance with a random walk)?

In Figure 2, rich landscape, experimental data, it seems that the X-shape is present but only within a small region around the center. In contrast, the model does not seem to present this feature. Is there a way the quantitatively assess the presence of the X-shape and then evaluate whether the model can produce to some extent also in the rich landscape?

In Figure 3C, for poor landscapes, the experimental data and the model seem to produce qualitatively different patterns. Can the authors quantitatively compare them and discuss this difference?

The presented model is composed of one particular strategy, which arbitrarily ranks payoffs above novelty in terms of priority. However, it would be interesting to study (1) whether other models like UCB (Auer et al 2002) or uncertainty bonus models (Daw O'Doherty et al 2006 Nature; Frank et al 2009 Nature Neuroscience; Wilson et al 2014 already cited; Gershman 2018 already cited; Cogliati-Dezza et al 2017 Scientific Reports) or similar models, which decide based on a linear combination of payoff and 'novelty' (or rather 'uncertainty' in their case, which is related in the present task); and (2) individual differences between subjects by testing whether different models (implementing different search strategies) are best suited to fit different subjects' data. As the authors mention in the discussion, their model roughly stands along the same line as models distinguishing directed and random exploration, because their model has a noise parameter and a novelty cue element. Nevertheless, the model arbitrarily sets a hierarchy between these two components while the above mentioned models use a weighting parameter between the two components, this parameter being a free parameter optimized per subject: different subjects give a different weight to novelty/uncertainty than others.

The model has 0 chance of revisiting a previously visited state, while this happens sometimes in subjects (14% of the time). Why not adding another noise parameter to allow for this? Would this second model perform better than the present one? During such a model comparison process, the authors should consider a way to penalize for model complexity, that is, the number of free parameters (Daw, N. D. (2011). Trial-by-trial data analysis using computational models. Decision making, affect, and learning: Attention and performance XXIII, 23(1).).

Figure 5B shows an example trajectory by a subject who could have explored a bit further (i.e., 5 steps more along the x axis) and would still have been able to get back to the optimum before the end of the experiment. As the authors note, this is an example of a suboptimal strategy. First, could the authors analyze the proportion of subjects that had a close to optimal strategy in this respect, and those who hadn't? In particular, I am wondering if the tendency to perform a suboptimal strategy with this respect is something stable within a given subject, or whether subjects sometimes performed an optimal one, and sometimes a suboptimal one. Second, it seems to that the risk to explore further after discovering an optimum (X_{best}) depends on the remaining number of rounds (T_{left}). Thus, maybe the analysis of the subjects' tendency (Figure 6) would be different if the authors could split the data into early trials ($T_{\text{left}} \geq 15$ to 20) and late trials ($T_{\text{left}} < 15$ to 20), or something like that. Figure 7B illustrates something similar to this intuition. By the way, the authors should specify what they mean by « exploration » and « exploitation » in Figure 7, which I think is simply moving away from X_{best} versus moving towards it. This should be clearly indicated to the reader. Anyway, What I mean is slightly different and is a prediction that the subjects should have the tendency to less perform the optimal strategy (thus explore) when $T_{\text{left}} \geq 15$, even when they still have time to do it and it would be optimal to do so, than when $T_{\text{left}} < 15$. It would be interesting if the authors could dig a bit more in this direction.

To me, Figure 6 shows a striking difference between experimental data and model for small payoffs values (upper-left corner): the experimental data show that the subjects tend to frequently

display high safety values (close to 1) in this area, while the model doesn't. Could the authors discuss this difference, try to explain it, and think of an alternative model which could account for this tendency?

I disagree with the authors that the model well captures the evolution of the average payoffs (figure 9) in the combined phase. One can see large series of rounds (especially between round 4 and round 15) where the model seems to perform significantly differently than the subjects. The authors should first perform some statistical tests to assess when these differences are significantly different. Moreover, the authors should be clearer here whether they used the version of the model with only 3 cues (from the exploration phase) or the one with 4 cues (from the exploitation phase). In addition, it is not satisfying to change the model between phases. A single model should explain behaviors in all phases. Could the authors test the model with 4 cues on the data of the exploration phase and discuss the results? Last but not least, this work would bring a stronger contribution to the field if the authors could compare their model with alternative models from the literature. I have cited a few examples above. I am sure the authors can find additional central models from their own field of research and compare all these models with quantitative methods. This way, the paper could show that model X cannot account for the data while model Y can. This would be useful in that the novel experimental data presented here could contribute to the refutation of existing models/theories. Without this, the model for the moment sounds a bit like an adhoc solution to account for the specific data gathered in this experiment.

The discussion section is bit short and does not bring enough perspective with respect to model predictions, what we learned from this study which was completely new compared to previous studies, what we learn about computational mechanisms used in existing models, etc. If the paper could include some model comparison, then the discussion could address the results of this model comparison and thus bring a wider perspective on the results.

Minor points

For clarity, in the legend of Figure 1, the authors should specify that the two example trajectories were obtained during the exploration phase.

Page 7, line 41, at this stage of reading, it is not clear what the authors mean by 'ranking cues according to their validity'. Could they please be more precise here?

Page 10, lines 49-50, the author should cite Figure 8 at the end of the following sentence: 'When looking at the density maps (figure X), the X-shaped exploration pattern is visible in the poor landscapes but not in the rich landscape, similar to the exploration phase.'

In terms of state of the art, I think that the tourist problem, with sharp transition between exploration and exploitation, as it is stated to present the contribution of the paper, is tightly related to foraging problems, which are studied by a literature which is overlooked here. The authors could benefit from looking at some of the related literature, both on the purely behavioural side, as well as on the behavioural neuroscience side: D. W. Stephens, J. R. Krebs, *Foraging Theory* (Princeton Univ. Press, Princeton, NJ, 1986). Hayden, B. Y., Pearson, J. M., & Platt, M. L. (2011). Neuronal basis of sequential foraging decisions in a patchy environment. *Nature neuroscience*, 14(7), 933. Kolling, N., Behrens, T. E., Mars, R. B., & Rushworth, M. F. (2012). Neural mechanisms of foraging. *Science*, 336(6077), 95-98.

Typos

Page 2, line 30: 'the separation of exploration and exploration' -> exploration and exploitation.

Page 6, line 49, 'an efficient strategy to extent one's exploration range' -> .. to extend ..

Author's Response to Decision Letter for (RSOS-190529.R0)

See Appendix A.

RSOS-190529.R1 (Revision)

Review form: Reviewer 2

Is the manuscript scientifically sound in its present form?

Yes

Are the interpretations and conclusions justified by the results?

Yes

Is the language acceptable?

Yes

Do you have any ethical concerns with this paper?

No

Have you any concerns about statistical analyses in this paper?

No

Recommendation?

Accept as is

Comments to the Author(s)

I think this version is substantially improved over the prior version. I also think the TTB heuristic provides an interesting thesis for thinking about search. I look forward to the future comparisons with area-restricted search in clumpy environments.

Review form: Reviewer 3 (Mehdi Khamassi)

Is the manuscript scientifically sound in its present form?

No

Are the interpretations and conclusions justified by the results?

No

Is the language acceptable?

Yes

Do you have any ethical concerns with this paper?

No

Have you any concerns about statistical analyses in this paper?

Yes

Recommendation?

Major revision is needed (please make suggestions in comments)

Comments to the Author(s)

The manuscript has been improved and now contains new simulations and analyses which bring further insight into the studied search behavior. I am glad that the model extension that I suggested (point 3.1.10) turns out to better fit the data and is now reflected in a new model with an additional parameter compared to the previous version of the manuscript.

It is a good point that the authors have uploaded their code, data, experimental protocol description and the instructions given to the subjects on github and zenodo. One minor comment here: On zenodo, the title of the referred manuscript is written 'Search as a simple the-best heuristic' instead of 'take-the-best'.

The manuscript is also strengthened by the addition of a new series of simulations testing another dimension: whether peaks are evenly distributed or clustered, and how this affects exploration strategies in the model. Nevertheless, since no experimental support is provided for these new model simulations, I think it is important to better emphasize these simulation results as model-driven experimental predictions, which could later be experimentally tested to further validate the model. Highlighting some predictions would definitively increase the value of the manuscript.

I am not fully satisfied by the answer to question 3.1.7 because the authors only indicate the complexity of each model (i.e., number of parameters), but then do not quantitatively include this in order to decide which model wins the model comparison. In the current manuscript, mean-square distances between model simulations and experimental data are compared for different models, but the models have different number of parameters. For instance, in Table 4, the returning model has better $\text{dist}_{\{\text{payoff}\}}$ than the simple returning model, a worst $\text{dist}_{\{\text{density}\}}$, and one more parameter. So who's the winning model? The authors should compare an aggregated score per model which takes into account penalties for model complexity (like Bayesian Information Criterion, Akaike Information Criterion, etc.).

The first line of Tables 3 and 4 gives exactly the same results: $\text{dist}_{\{\text{payoff}\}} = 1.21$; $\text{dist}_{\{\text{density}\}} = 2.18$. Is there a mistake here?

I am not fully satisfied by the answer to question 3.1.1. The authors found that TTB revisits previously seen states 14% of the time, while a random walk does it 57% of the time. I am a bit surprised by this very high rate for the random walk. I am pretty sure there is a lot of variability in this rate when launching several simulations of the same random walk model. Could the authors please indicate what is the confidence interval for each model, and then compare the results with a statistical tests (ANOVA if the requirements are met, or Mann-Whitney otherwise)?

Given the unsatisfying performance of the TTB model in the poor environments, is it possible that subjects are following a different strategy than TTB or random walk in this case? To help answer this question, it would be useful to present additional model comparison in poor environments only, and in rich environments only.

The additional analyses performed by the authors in response to question 3.1.9, where they split the experiment into three equal parts, show a tendency in the subjects which is not captured by the model. This should be added to the manuscript, at least to the supplementary material, and should be mentioned in the discussion.

Typo: Line 278: return the their best solution -> return to their best solution.

Decision letter (RSOS-190529.R1)

22-Aug-2019

Dear Mr Yahosseini:

Manuscript ID RSOS-190529.R1 entitled "Search as a simple take-the-best heuristic" which you submitted to Royal Society Open Science, has been reviewed. The comments of the reviewer(s) are included at the bottom of this letter.

Please submit a copy of your revised paper before 14-Sep-2019. Please note that the revision deadline will expire at 00.00am on this date. If we do not hear from you within this time then it will be assumed that the paper has been withdrawn. In exceptional circumstances, extensions may be possible if agreed with the Editorial Office in advance. We do not allow multiple rounds of revision so we urge you to make every effort to fully address all of the comments at this stage. If deemed necessary by the Editors, your manuscript will be sent back to one or more of the original reviewers for assessment. If the original reviewers are not available we may invite new reviewers.

- Ethics statement

- Data accessibility

- Competing interests

- Authors' contributions

- Acknowledgements

- Funding statement

on behalf of Dr Narayanan Srinivasan (Associate Editor) and Essi Viding (Subject Editor)
openscience@royalsociety.org

Associate Editor Comments to Author (Dr Narayanan Srinivasan):

Two reviewers have now commented on the paper. While reviewer 1 is fully satisfied with your revision, reviewer 2 still has some issues that need to be addressed. You can address these issues and submit a revision with response to Reviewer 2's comments.

Reviewer comments to Author:
Reviewer: 2

I think this version is substantially improved over the prior version. I also think the TTB heuristic provides an interesting thesis for thinking about search. I look forward to the future comparisons with area-restricted search in clumpy environments.

Reviewer: 3

Comments to the Author(s)

The manuscript has been improved and now contains new simulations and analyses which bring further insight into the studied search behavior. I am glad that the model extension that I suggested (point 3.1.10) turns out to better fit the data and is now reflected in a new model with an additional parameter compared to the previous version of the manuscript.

It is a good point that the authors have uploaded their code, data, experimental protocol description and the instructions given to the subjects on github and zenodo. One minor comment here: On zenodo, the title of the referred manuscript is written 'Search as a simple the-best heuristic' instead of 'take-the-best'.

The manuscript is also strengthened by the addition of a new series of simulations testing another dimension: whether peaks are evenly distributed or clustered, and how this affects exploration strategies in the model. Nevertheless, since no experimental support is provided for these new model simulations, I think it is important to better emphasize these simulation results as model-driven experimental predictions, which could later be experimentally tested to further validate the model. Highlighting some predictions would definitely increase the value of the manuscript.

I am not fully satisfied by the answer to question 3.1.7 because the authors only indicate the complexity of each model (i.e., number of parameters), but then do not quantitatively include this in order to decide which model wins the model comparison. In the current manuscript, mean-square distances between model simulations and experimental data are compared for different models, but the models have different number of parameters. For instance, in Table 4, the returning model has better $\text{dist}_{\{\text{payoff}\}}$ than the simple returning model, a worst $\text{dist}_{\{\text{density}\}}$, and one more parameter. So who's the winning model? The authors should compare an aggregated score per model which takes into account penalties for model complexity (like Bayesian Information Criterion, Akaike Information Criterion, etc.).

The first line of Tables 3 and 4 gives exactly the same results: $\text{dist}_{\{\text{payoff}\}} = 1.21$; $\text{dist}_{\{\text{density}\}} = 2.18$. Is there a mistake here?

I am not fully satisfied by the answer to question 3.1.1. The authors found that TTB revisits previously seen states 14% of the time, while a random walk does it 57% of the time. I am a bit surprised by this very high rate for the random walk. I am pretty sure there is a lot of variability in this rate when launching several simulations of the same random walk model. Could the authors please indicate what is the confidence interval for each model, and then compare the results with a statistical tests (ANOVA if the requirements are met, or Mann-Whitney otherwise)?

Given the unsatisfying performance of the TTB model in the poor environments, is it possible that subjects are following a different strategy than TTB or random walk in this case? To help answer this question, it would be useful to present additional model comparison in poor environments only, and in rich environments only.

The additional analyses performed by the authors in response to question 3.1.9, where they split the experiment into three equal parts, show a tendency in the subjects which is not captured by the model. This should be added to the manuscript, at least to the supplementary material, and should be mentioned in the discussion.

Typo: Line 278: return the their best solution -> return to their best solution.

Author's Response to Decision Letter for (RSOS-190529.R1)

See Appendix B.

RSOS-190529.R2 (Revision)

Review form: Reviewer 3 (Mehdi Khamassi)

Is the manuscript scientifically sound in its present form?

Yes

Are the interpretations and conclusions justified by the results?

Yes

Is the language acceptable?

Yes

Do you have any ethical concerns with this paper?

No

Have you any concerns about statistical analyses in this paper?

No

Recommendation?

Accept as is

Comments to the Author(s)

The authors have considerably improved their manuscript. Congratulations for this work!

Decision letter (RSOS-190529.R2)

28-Sep-2019

Dear Mr Yahosseini,

I am pleased to inform you that your manuscript entitled "Search as a simple take-the-best heuristic" is now accepted for publication in Royal Society Open Science.

Please ensure you email the editorial office with:

1. Individual figure files;
2. Individual table files;
3. A file containing captions for the above.

on behalf of Dr Narayanan Srinivasan (Associate Editor) and Essi Viding (Subject Editor)
openscience@royalsociety.org

Associate Editor Comments to Author (Dr Narayanan Srinivasan):

Associate Editor: 1

Comments to the Author:

The reviewer who had significant comments have now reviewed the paper and is satisfied with the revisions made.

Reviewer comments to Author:

Reviewer: 3

Comments to the Author(s)

The authors have considerably improved their manuscript. Congratulations for this work!

Appendix A

Berlin, July 11, 2019

Dear Dr Srinivasan,

We have revised our manuscript in accordance with the reviewers' comments. The main changes are:

1. Model extension: We extended our heuristic TTB model with a new parameter: the *minimal acceptable payoff*. We show that this parameter can correct the differences between the model predictions and the experimental data that the reviewer #3 has highlighted.
2. Model comparison: As suggested by two reviewers, we extended our manuscript to include a comparison between our model and alternative ones.
3. Individual differences: We conducted additional analyses to describe individual differences in terms of safety level and cue usage.
4. Landscape structure: To better understand the influence of the solution space on the model predictions, we ran additional simulations varying the patchiness of landscapes (i.e. the extent to which peaks are clustered in one region of the landscape).
5. Clarifications and additional analysis: In accordance with the reviewers comments, we clarified some parts of the manuscript and added new analyses to support our findings.

Detailed responses to the referees can be found in the remainder of this letter. We appreciate greatly the efforts of the reviewers, and hope that our revised manuscript meets the standards of Royal Society Open Science.

We look forward to the evaluation of our revised manuscript.

Yours sincerely,

The authors

Kyanoush Seyed Yahosseini and Mehdi Moussaïd

Reply to the reviewers

Reviewer 1

1.1 The manuscript would benefit from more clearly fleshing out what specific (real-world) choice problems come with ‘gradual exploration’ and a ‘temporal separation between exploration and exploitation’. Adding this information beyond merely stating that these choice situations occur in visual search, spatial search and problem-solving situations would make the manuscript more appealing to a broader audience.

Response: *We are now giving more detailed examples of search problems that can be described by gradual exploration and separation between exploration and exploitation (page 2, line 32).*

1.2 Please add more information regarding the instructions that participants received and what training they completed. What was the “cover story” for the search problem that participants completed?

Response: *The participants did not receive a particular cover story, but were told to either search for the highest payoff or end at the highest payoff (depending on the experimental phase). Participants completed a training at the beginning of each of the three experimental phases. The training included: 1) information about the goal of the search and the size of the landscape, 2) information about the nine possible moving options, and 3) a fully functional exercise in a randomly generated landscape, for which the responses were not recorded. All introductory messages shown to the participants can be accessed at https://github.com/cuehs/search_heuristic/blob/master/experiment/webapp/js/text.js (in german).*

We have added a figure illustrating the experimental interface in the supplementary materials. The description of the experimental setup has also been expanded in the manuscript’s methods section (page 3 line 56 and page 4, line 102).

1.3. Many other choice problems that the authors describe in their paper are imbedded in very specific choice environments. Often, the statistical regularities of these choice environments tell us a lot about what participants should or should not do. While the choice environment is somewhat specified in the current study, I could not fully make sense of what drove the distribution of normalized payoffs that was used. Is there a particular reason that a more or less evenly dispersed distribution was chosen for the poor and rich landscape? Would the obtained results be the same if these payoff distributions were to be changed—say into a random distribution of payoff distributions with clusters/patches of resources?

Response: *Search environments can vary along various dimensions. One such dimension is the difficulty of the task, which we manipulated in the manuscript by varying the number of peaks in the landscape. Nevertheless, other aspects of the search environment could be varied as well, such as the peaks size or height. Following the reviewer’s suggestion, we*

have added another dimension to our analysis, namely, the peaks location: peaks can be either dispersed evenly over the entire landscape, or clustered in one region of the landscape, creating “patchy” environments. We ran additional simulations showing the dynamics predicted by the model when varying the patchiness of the landscapes and added the results in the supplementary materials. We also refer to these new results in the discussion (page 13, line 327)

Figure 1.3. An example of simulated search trajectory in a patchy landscape, where all solutions are clustered in the same area. The red dots indicate the participant's trajectory, and the colour-coding indicates the normalised payoff associated to each position (x,y) .

Reviewer 2

2.1 Search paradigm and environment structure: The environments explored and the paradigm used to study search represent a small subset of what many people might call search. Both the poor and rich environment in Figure 1 are fairly sparse and clumps in these environments seem to rarely represent more than 9 pixels. This means it will rarely be meaningful to return to a previously encountered cluster, and clusters can be harvested fairly straightforwardly upon encounter. The TTB heuristic captures this harvesting phase and the movement away from previously visited solutions. But moving away from previously visited cells (a combination of the non-visited and novelty cue) and harvesting those you encounter is not what I would call search in any but the most trivial sense. This would not capture one of the most commonly observed search patterns area-concentrated search, because it would not move back to previously visited clusters after it left them. The TTB search process has no memory, much like a Lévy flight, which throws this heuristic into the cognitive capacities of a bacterium. BUT, a bacterium has a short memory, and will turn around if it starts to move down a resource gradient. TTB has no such memory. However, the search environment investigated doesn't have clusters sufficiently large to engage patch or cluster based strategies, making them perhaps unrealistic even for bacteria.

I would suggest including a search environment with larger clusters and without the ability to see neighboring cells. The problem is that with this latter constraint (the inability to see neighboring cues), the TTB heuristic is not realistic. This tells us something about the TTB heuristic. It is not a search strategy, but a harvest strategy. TTB can only take what it has already found. The question of search is really 'how does one look?'

***Response:** In fact, our modeling framework can be updated to account for the situations the reviewer describes. The case where an individual could not see (nor sense) the payoff values of the neighboring cells can be easily produced by turning off the payoff cue in the exploration process. The simulated agent would then navigate only by searching for novel and non-visited cells (see the "blind search" model in the new 'model comparison' section of the manuscript). The exploitation process remains unchanged: whenever time is running out, the agent would gradually move back to the best discovered solution.*

In fact, numerous variations of the model can be produced, simply by turning on and off certain cues (as we describe in the next point). We have now implemented this aspect in the new 'model comparison' section, and discuss this point in the light of additional references.

Besides, we have implemented search environments with larger clusters as suggested by the reviewer (the so-called "patchy" landscapes, see also our answer to the comment 1.3 of reviewer #1), on which we simulated different model variations (see the 'model comparison' section at page 11 line 252, the discussion about clustered landscapes at page 13 line 327, and supplementary results).

2.2. Why not compete heuristics? The heuristic described is straightforward. But is it the only possibility? I think the work would be improved by competing heuristics against one another. This will also tease apart the differences between different environments. For example, consider a 'harvest' and 'explore' strategy. This strategy takes the best, after if it finds it,

same as the payoff cue, when it is visible in the surrounding cells. Otherwise, it always moves away from the last visited-cue. One could add that it moves diagonally away from the last visited cue. This last constraint though is an artefact of having space distributed as a pixel array. Alternatively, one could try another heuristic, if one finds oneself in a field of view without resources immediately after being in a field of view with resources, then turn around. This is the bacterium's strategy and it makes sense in a clumpy world (see for example Pacheco-Cobos et al. 2019 and Hutchinson et al 2012 cited therein). One could also develop some Levy walk like heuristics.

Response: *We have taken into account the suggestion of the reviewer and added a new section to the manuscript to compare our model to alternative ones (page 11). The models suggested by the reviewer were included in the comparison:*

- *A random search model (or Brownian motion) can be produced by turning off all cues in our TTB framework. The simulated agent thus explores the landscape completely randomly.*
- *A “blind search” model ignoring the payoff values of adjunct solutions, but moving towards the most novel cells – similar to the bacterium's model described by the reviewer.*
- *A simple “hill climbing” model where only the payoff cue is taken into account.*
- *A probabilistic model based on a linear combination of payoff and novelty.*

In practice, even a Levy flight model could be derived from our TTB framework by ignoring the payoff cue for a certain number of steps taken from a long-tailed distribution, causing straight movement towards one direction. We did not, however, implement this variation in our model comparison to avoid injecting too much complexity in the process.

2.3 I think the analysis is good and I appreciate the details. I would like to know how often the heuristic (and its competitors) make a choice consistent with the observed behaviour of the participants.

Response: *Figure 3 shows the proportion of actual moves that the participants made and the corresponding model predictions, depending on the considered cue. Overall, the observed decisions are in line with the model predictions, except for the novelty cue in poor landscapes, which are less frequent than predicted but remain nevertheless more frequent than what a random search model predicts. Overall the three cues explain 68.8 % of all the observed decisions. We have updated the caption of figure 3 accordingly.*

2.4 Is $\epsilon = 0.17$ true for all environments?

Response: *Yes indeed, we fitted the noise parameter ϵ once, in the exploration phase. To avoid overfitting, ϵ was subsequently kept constant at .17. That is, in all phases there is a .17 chance that a random option not supported by the cues is chosen. We have now clarified this point in the manuscript (page 6, line 155).*

Reviewer 3

3.1.1. Could the authors present some statistical test to assess the significance of the observation that ‘In both environments, participants avoided revisiting a previously visited solutions (on average 14% of the solutions were visited more than once).’ (page 5, lines 56-58)? In particular, is this percentage significantly higher than what can be obtained by chance (for instance with a random walk)?

Response: *To answer this comment, we have implemented a random search model that could be used as a benchmark for evaluating our simulation results. In that model, the direction of the movement is chosen at random, irrespective of any cue value. Simulations show that this model predicts a ratio of 57% of solutions visited more than once, which is considerably higher than the 14% observed in our data and predicted by our TTB model. We have now added this information in the manuscript (page 4, line 117.).*

3.1.2. In Figure 2, rich landscape, experimental data, it seems that the X-shape is present but only within a small region around the center. In contrast, the model does not seem to present this feature. Is there a way the quantitatively assess the presence of the X-shape and then evaluate whether the model can produce to some extent also in the rich landscape?

Response: *We have introduced a formal measure of similarity $dist_{density}$ to calculate the average difference in density between two maps. This measure allows us now to quantify the similarity between observed and simulated maps, which we use in the new ‘model comparison’ section (see response to 3.1.4. below). In addition, one can specifically assess the presence of the X-shape, by comparing the density maps along the diagonal lines to an artificially-generated X pattern (i.e., an empty map with filled diagonal lines). This method confirms that the pattern is more marked in the poor landscapes than in the rich ones ($dist_{density}$ between the experimental data and the artificial X-shape is 79.9 and 88.1 for the poor and rich landscapes, respectively). In the model simulations, the X pattern is indeed less present than observed in the experimental data, especially on rich landscapes ($dist_{density}$ between the model’s prediction and the artificial X-shape is 81.5 and 95.1 for the poor and rich landscapes, respectively). Nevertheless, the model remains closer to observed density maps than the other models that we tested (see point 3.1.4).*

3.1.3. In Figure 3C, for poor landscapes, the experimental data and the model seem to produce qualitatively different patterns. Can the authors quantitatively compare them and discuss this difference?

Response: *Indeed, in poor landscapes participants tend to rely less on the similarity cue than what is predicted by our model (the observed mean rate of decisions based on that cue is 57%). It remains, nevertheless, considerably higher than what a random model predicts (40%) suggesting that individuals do exhibit a bias towards revealing novel solutions, although not as systematically as the model predicts. The agreement is much better in the*

rich landscape condition. We have added this note to the manuscript (page 6 caption figure 3)

3.1.4. The presented model is composed of one particular strategy, which arbitrarily ranks payoffs above novelty in terms of priority. However, it would be interesting to study (1) whether other models like UCB (Auer et al 2002) or uncertainty bonus models (Daw O'Doherty et al 2006 Nature; Frank et al 2009 Nature Neuroscience; Wilson et al 2014 already cited; Gershman 2018 already cited; Cogliati-Dezza et al 2017 Scientific Reports) or similar models, which decide based on a linear combination of payoff and 'novelty' (or rather 'uncertainty' in their case, which is related in the present task);

Response: *The challenge is that these models cannot be directly applied to our experimental paradigm. In fact, they interweave exploration and exploitation and are difficult to apply for problems where individuals need to decide when to stop exploring and start exploiting. This concerns both the models describing search behavior in the multi-armed bandits (Auer et al 2002; Daw O'Doherty et al 2006 Nature) and in the horizon tasks (Wilson et al 2014; Cogliati-Dezza et al 2017). In contrast, our model has two distinct components: (1) how the individual explores, and (2) when the individual stops the exploration and starts the exploitation. To overcome this limitation, we kept the exploitation mechanism unchanged and tested five different models for the exploration phase.*

We included a variation of the probabilistic model inspired by the mechanism that Wilson and colleagues have proposed. That model assumes that the agents choose probabilistically between using the payoff or the visibility cue. Formally, we fitted a logistic regression to the probability of relying on the payoff cue as a function of the highest neighboring payoff.

The comparison between our TTB model and other ones (including the probabilistic model) is now presented in a new section of the manuscript (see page 11).

3.1.5. (2) individual differences between subjects by testing whether different models (implementing different search strategies) are best suited to fit different subjects' data. As the authors mention in the discussion, their model roughly stands along the same line as models distinguishing directed and random exploration, because their model has a noise parameter and a novelty cue element. Nevertheless, the model arbitrarily sets a hierarchy between these two components while the above mentioned models use a weighting parameter between the two components, this parameter being a free parameter optimized per subject: different subjects give a different weight to novelty/uncertainty than others.

Response: *We have conducted a subject-based analysis to evaluate possible individual differences in the usage of the different cues. For that, we calculated a linear mixed effect model expanding on the probabilistic model introduced in 3.1.4, and added participants as a random effect and intercept. Our results, indeed, revealed individual differences in the way participants threat cues ($\chi^2 = 359$, $p < 10^{-10}$), but these differences were relatively small (e.g. the inflection points of all fitted curves are between a most rewarding payoff of 5 and 8, see also figure 3.1.5.). This information has now been added to the supplementary materials (also see response to comment 3.1.8)*

Figure 3.1.5. Relationship between the payoff of the most-rewarding neighbouring solution and the probability to rely on the payoff cue instead of the visibility cue. Each dot indicates one landscape. Each black line is the the best logistic function describing an individual's behavior (mixed effect model with participants as a random effect). The red line indicates the best logistic regression for the whole group, that is, no using the individual participant as a random effect.

3.1.6. The model has 0 chance of revisiting a previously visited state, while this happens sometimes in subjects (14% of the time). Why not adding another noise parameter to allow for this? Would this second model perform better than the present one?

Response: In the previous version of the manuscript, we mistakenly wrote that the noise parameter ϵ was the probability to make a random choice between all non-visited options. This is not correct: our noise parameter is defined as the probability to make a random choice between **all available options**.

Thus, there exists a non-zero chance that a simulated agent would revisit a previously explored solution. In fact, figure 3A) shows a similar rate of movement towards already visited solutions between simulations and experimental data. We have updated the manuscript accordingly (page 6 line 150)

3.1.7. During such a model comparison process, the authors should consider a way to penalize for model complexity, that is, the number of free parameters (Daw, N. D. (2011). Trial-by-trial data analysis using computational models. Decision making, affect, and learning: Attention and performance XXIII, 23(1).).

Response: In the new model comparison section, we now report the number of free parameters and number of cues used by each model, as an indicator of complexity (see page 11).

3.1.8. First, could the authors analyze the proportion of subjects that had a close to optimal strategy in this respect, and those who hadn't? In particular, I am wondering if the tendency to perform a suboptimal strategy with this respect is something stable within a given subject, or whether subjects sometimes performed an optimal one, and sometimes a suboptimal one.

Response: The optimality of the strategy is determined by one parameter in our model: the safety level. For a safety level of 1, the individuals use the maximum available time for exploration before returning to the best solution, whereas a suboptimal strategy would consist in returning earlier (i.e. for a safety level < 1).

We therefore assessed the stability of the safety levels for each individual participant between the exploitation phase and the combined phase of our experiment. The results indicate a linear correlation between the safety level displayed in the two phases (see figure 3.1.8., Pearson's correlation = .42, $df=47$, $p < 0.003$). The standard error also indicates that the individual's safety level in the combined phase is surprisingly stable.

Because the safety level reflects a participant's propensity for taking risks, our finding is consistent with research suggesting that risk preferences tend to stay constant over time and tasks (Frey et al. 2017. "Risk Preference Shares the Psychometric Structure of Major Psychological Traits."). We now refer to this finding in the manuscript (page 10 line 245 and page 13 line 319) and have added a section about individual differences in the supplementary materials.

Figure 3.1.8: Safety levels displayed in two different phases of the experiment. Each point indicates the average safety level of one participant in the exploitation and the combined phase. The error bars indicate the standard errors in the corresponding phase.

3.1.9. Second, it seems that the risk to explore further after discovering an optimum (X_{best}) depends on the remaining number of rounds (T_{left}). Thus, maybe the analysis of the subjects' tendency (Figure 6) would be different if the authors could split the data into early trials ($T_{\text{left}} \geq 15$ to 20) and late trials ($T_{\text{left}} < 15$ to 20), or something like that. Figure 7B illustrates something similar to this intuition. [moved to 3.2.1] Anyway, What I mean is slightly different and is a prediction that the subjects should have the tendency to less perform the optimal strategy (thus explore) when $T_{\text{left}} \geq 15$, even when they still

have time to do it and it would be optimal to do so, than when $T_{\text{left}} < 15$. It would be interesting if the authors could dig a bit more in this direction.

Response: *The question that the reviewer is asking, is whether the safety level exhibited by the participants depends on the remaining number of rounds T_{left} . To address this issue, we split the data into three equal parts based on the time when the best solution P_{best} is found: early, intermediate, and late, corresponding to rounds 1-10, 11-20, and 21-30. As the reviewer anticipated, T_{left} influences the safety level of the participants (see figure 3.1.9A). Participants tend to display a higher safety level when a good solution is found earlier than when it is found later – a trend that is not captured by our model (see figure 3.1.9B). It would, in principle, be possible to implement such a dependency in the model by decreasing the safety level proportionally to T_{left} . This, however, would require the use of another free parameter and would considerably increase the model complexity and accessibility – without adding much explanatory power to it. We therefore chose not to implement this dependency.*

Figure 3.1.9. Safety level S as a function of the payoff P_{best} A) as observed in the experimental data, and B) as obtained from numerical simulations. Early, intermediate and late refer to the round where P_{best} is discovered, that is, round 1-10, 11-20, and 21-30 respectively.

3.1.10. To me, Figure 6 shows a striking difference between experimental data and model for small payoff values (upper-left corner): the experimental data show that the subjects tend to frequently display high safety values (close to 1) in this area, while the model doesn't. Could the authors discuss this difference, try to explain it, and think of an alternative model which could account for this tendency?

Response: *This difference can be explained by the fact that very low payoffs are often considered “not good enough” by participants who do not return to them (even when no*

better solution has been found). In contrast, our simulated agents have no lower bound aspiration level.

This behavior can be implemented by means of a new parameter I , indicating the extent to which individuals ignore low payoff peaks. After fitting I to the experimental data, a new series of simulations show that the updated model is in better agreement with the data: individuals display high safety levels for low payoff values (see figure 6). Interestingly, this new parameter also improved the agreement with the data at the payoff level (i.e. in figure 9).

We have therefore extended our model with this additional parameter. The manuscript has been updated to reflect these changes (page 7 line 186, page 8, page 11 line 238, and figures 6, 7, 8, and 9)

3.1.11. I disagree with the authors that the model well captures the evolution of the average payoffs (figure 9) in the combined phase. One can see large series of rounds (especially between round 4 and round 15) where the model seems to perform significantly differently than the subjects. The authors should first perform some statistical tests to assess when these differences are significantly different.

Response: *The new parameter that was added to the model in response to the previous comment (see 3.1.10) has also improved the quality of the simulation results in Figure 9. In fact, the discrepancy between data and simulations that the reviewer has highlighted was due to the same issue that caused the mismatch in figure 6. The manuscript has been updated accordingly.*

3.1.12. Moreover, the authors should be clearer here whether they used the version of the model with only 3 cues (from the exploration phase) or the one with 4 cues (from the exploitation phase). In addition, it is not satisfying to change the model between phases. A single model should explain behaviors in all phases. Could the authors test the model with 4 cues on the data of the exploration phase and discuss the results?

Response: *This is an important point. The full model, introduced in the combined phase, is the final model that we would like to propose in our manuscript. It is based on two building blocks: the exploration behaviour and the exploitation behaviour. These two blocks were developed separately in the first two experimental phases, and later merged to produce the full model.*

Therefore, we have followed a bottom-up construction procedure: We first elaborated the exploration model (in the exploration phase), then developed the exploitation behaviour (in the exploitation phase) and finally combined the two to produce the full model. We have now clarified this methodological aspect in the manuscript (page 9 line 233) .

3.1.13. Last but not least, this work would bring a stronger contribution to the field if the authors could compare their model with alternative models from the literature. I have cited a few examples above. I am sure the authors can find additional central models from their own field of research and compare all these models with quantitative methods. This way, the paper could show that model X cannot account for the data while model Y can. This would be useful in that the novel experimental data presented here could contribute to the

refutation of existing models/theories. Without this, the model for the moment sounds a bit like an adhoc solution to account for the specific data gathered in this experiment.

Response: *Absolutely. We have now added a new 'model comparison' section to the manuscript, which includes some of the models suggested by the reviewers 2 and 3 (see point 3.1.4.)*

3.1.14. The discussion section is a bit short and does not bring enough perspective with respect to model predictions, what we learned from this study which was completely new compared to previous studies, what we learn about computational mechanisms used in existing models, etc. If the paper could include some model comparison, then the discussion could address the results of this model comparison and thus bring a wider perspective on the results.

Response: *The discussion has been extended and now refers to a number of issues that have been raised by the reviewers, including inter-individual differences, variations of the search environment, and the results of the model comparison.*

Minor points

3.2.1 By the way, the authors should specify what they mean by « exploration » and « exploitation » in Figure 7, which I think is simply moving away from X_{best} versus moving towards it. This should be clearly indicated to the reader.

For clarity, in the legend of Figure 1, the authors should specify that the two example trajectories were obtained during the exploration phase.

Page 7, line 41, at this stage of reading, it is not clear what the authors mean by 'ranking cues according to their validity'. Could they please be more precise here?

Page 10, lines 49-50, the author should cite Figure 8 at the end of the following sentence: 'When looking at the density maps (figure X), the X-shaped exploration pattern is visible in the poor landscapes but not in the rich landscape, similar to the exploration phase.'

Response: *We have now made the suggested changes to the manuscript.*

3.2.3 In terms of state of the art, I think that the tourist problem, with sharp transition between exploration and exploitation, as it is stated to present the contribution of the paper, is tightly related to foraging problems, which are studied by a literature which is overlooked here. The authors could benefit from looking at some of the related literature, both on the purely behavioural side, as well as on the behavioural neuroscience side: D. W. Stephens, J. R. Krebs, Foraging Theory (Princeton Univ. Press, Princeton, NJ, 1986). Hayden, B. Y., Pearson, J. M., & Platt, M. L. (2011). Neuronal basis of sequential foraging decisions in a patchy environment. *Nature neuroscience*, 14(7), 933. Kolling, N., Behrens, T. E., Mars, R. B., & Rushworth, M. F. (2012). Neural mechanisms of foraging. *Science*, 336(6077), 95-98.

Response: *We thank the reviewer for recommending these publications. We have now added the case of foraging as an example of search problems with gradual exploration and a temporal separation between exploration and exploitation.*

3.2.4 Page 2, line 30: 'the separation of exploration and exploration' -> exploration and exploitation. Page 6, line 49, 'an efficient strategy to extent one's exploration range' -> .. to extend ..

Response: *We fixed the spelling mistakes and conducted an additional proofreading check.*

Appendix B

Berlin, September 6, 2019

Dear Dr Srinivasan,

We have revised our manuscript in accordance with the comments of the reviewer #3. The two main changes are:

1. Cross-validated model comparison: To avoid potential overfitting, we now use cross validation in our model comparison section.
2. Clarifications and additional analysis: In response to the reviewer's comments, we have clarified some parts of the manuscript and added new analyses to support our findings.

Detailed responses to the reviewer #3 can be found in the remainder of this letter. We greatly appreciate the efforts of the three reviewers, and hope that our revised manuscript meets the standards of Royal Society Open Science.

We look forward to the evaluation of our revised manuscript.

Yours sincerely,

The authors

Kyanoush Seyed Yahosseini and Mehdi Moussaïd

Reply to the reviewers

Reviewer 3

3.1 The manuscript is also strengthened by the addition of a new series of simulations testing another dimension: whether peaks are evenly distributed or clustered, and how this affects exploration strategies in the model. Nevertheless, since no experimental support is provided for these new model simulations, I think it is important to better emphasize these simulation results as model-driven experimental predictions, which could later be experimentally tested to further validate the model. Highlighting some predictions would definitively increase the value of the manuscript.

Response: *Our manuscript does not, indeed, include an experimental validation of the search behavior in patchy environments. We have now clarified this fact in the manuscript and suggest future research to test the model's predictions (page 13 line 341). In addition, the predicted patterns for patchy environments have been included in the supplementary materials.*

3.2. I am not fully satisfied by the answer to question 3.1.7 because the authors only indicate the complexity of each model (i.e., number of parameters), but then do not quantitatively include this in order to decide which model wins the model comparison. In the current manuscript, mean-square distances between model simulations and experimental data are compared for different models, but the models have different number of parameters. For instance, in Table 4, the returning model has better $\text{dist}_{\{\text{payoff}\}}$ than the simple returning model, a worst $\text{dist}_{\{\text{density}\}}$, and one more parameter. So who's the winning model? The authors should compare an aggregated score per model which takes into account penalties for model complexity (like Bayesian Information Criterion, Akaike Information Criterion, etc.).

Response: *Models with more parameters are more flexible when fitting data, but too many parameters increase the risk of overfitting and are less generalizable when predicting data. Akaike Information Criterion (AIC) and Bayesian Information Criterion (BIC) are two commonly-used criteria to compare models with different number of parameters. They consist in evaluating the models fit by means of a likelihood function while at the same time penalizing their number of free parameters. Our model, however, has no straightforward way of calculating such a likelihood function.*

Therefore, we have used another method to compare models with different number of parameters and avoid overfitting (i.e., the purpose of AIC and BIC): k-fold cross-validation, with $k=5$ (see page 15 in "Daw, N. D. (2011). Trial-by-trial data analysis using computational models"). Here, different parts of the dataset are used to fit the model parameters (the training set) and to measure the prediction accuracy (the test set). We hence divided our data in 5 subsamples of the same size: One subsample as the test set, and the remaining 4

as training sets. The cross-validation process is then repeated 5 times, where each of the 5 subsamples is used exactly once as the test data to calculate *dist_payoff* and *dist_density*. The results are then averaged to produce a single estimation.

The new cross-validated measurements of *dist_payoff* and *dist_density* are hence out of sample **predictions** and automatically incorporate a measure to deal with overfitting. As a side effect, this approach resolved disagreements between *dist_density*, and *dist_payoff*. We have now updated the manuscript to reflect these changes (page 10, line 270, page 12, line 286, and table 3 and 4).

3.7 The first line of Tables 3 and 4 gives exactly the same results: $\text{dist}_{\{\text{payoff}\}} = 1.21$; $\text{dist}_{\{\text{density}\}} = 2.18$. Is there a mistake here?

Response: *The first line in tables 3 and 4 both refer to the same model : “take-the-best model” for exploration and “returning model” for exploitation. Hence, the measurements $\text{dist}_{\text{payoff}}$ and $\text{dist}_{\text{density}}$ are indeed the same. We have emphasized this in the captions of tables 3 and 4.*

3.3. I am not fully satisfied by the answer to question 3.1.1. The authors found that TTB revisits previously seen states 14% of the time, while a random walk does it 57% of the time. I am a bit surprised by this very high rate for the random walk. I am pretty sure there is a lot of variability in this rate when launching several simulations of the same random walk model. Could the authors please indicate what is the confidence interval for each model, and then compare the results with a statistical tests (ANOVA if the requirements are met, or Mann-Whitney otherwise)?

Response: *As a random walk has no memory of previously visited solutions, the frequency of visiting the same solutions several times is considerably high. The density map resulting from a random walk covers a substantially smaller area than all the other proposed exploration methods (see figure A3 in the supplementary materials), indicating a high revisit rate.*

As the reviewer foresaw, the standard deviation of this rate is indeed pretty high (.10). Nevertheless, the rate remains significantly higher than the experimental data, as indicated by a Mann-Whitney-Wilcoxon test ($W = 149730$, $p\text{-value} < 2.2e-16$). We have added information about the standard deviations to the manuscript (page 4 line 118).

3.4. Given the unsatisfactory performance of the TTB model in the poor environments, is it possible that subjects are following a different strategy than TTB or random walk in this case? To help answer this question, it would be useful to present additional model comparison in poor environments only, and in rich environments only.

Response: *Our data indicate that the performance of individuals in poor environments is lower than in rich environments (mean normalized performance in the combined phase is .36 and .24 in rich and poor landscapes, respectively). This decay was expected and is not surprising: finding a good solutions in poor environments is more difficult than in rich ones, irrespective of the search strategy. Our model captures this decay of performance pretty*

well. The quality of the fit (not cross validated) in poor environments ($P_{\text{payoff}}=1.26$ and $P_{\text{density}}=1.59$) is, indeed, comparable to the fit in rich environments ($P_{\text{payoff}}=1.18$ and $P_{\text{density}}=1.08$). In other words, the unsatisfactory performance is simply due to the task being more difficult, which applies to experimental participants as well as simulated agents. Figure 9 confirms that experimental data and numerical simulations are undergoing the same performance decay in poor environments .

3.5.The additional analyses performed by the authors in response to question 3.1.9, where they split the experiment into three equal parts, show a tendency in the subjects which is not captured by the model. This should be added to the manuscript, at least to the supplementary material, and should be mentioned in the discussion.

Response: *Indeed, some aspects of the search patterns are not captured by our model, such as the time dependency of S . A model cannot naturally capture every single facet of the observed behaviour, but it is nevertheless important to stress its limitations for stimulating future research. We have now mentioned this inadequacy in the manuscript (page 12 line 287) and added the suggested figure in the supplementary materials.*

3.6 It is a good point that the authors have uploaded their code, data, experimental protocol description and the instructions given to the subjects on github and zenodo. One minor comment here: On zenodo, the title of the referred manuscript is written 'Search as a simple the-best heuristic' instead of 'take-the-best'.

Typo: Line 278: return the their best solution -> return to their best solution.

Response: *We have fixed the spelling mistakes and updated the GitHub repository and zendodo to correct the mistake highlighted by the reviewer.*